# G4Seg: Generation for Inexact Segmentation Refinement with Diffusion Models

## Abstract

This paper considers the problem of utilizing a large-scale text-to-image diffusion model to tackle the challenging Inexact Segmentation (IS) task. Unlike traditional approaches that rely heavily on discriminative-model-based paradigms or dense visual representations derived from internal attention mechanisms, our method focuses on the intrinsic generative priors in Stable Diffusion (SD). Specifically, we exploit the pattern discrepancies between original images and mask-conditional generated images to facilitate a coarse-to-fine segmentation refinement by establishing a semantic correspondence alignment and updating the foreground probability. Comprehensive quantitative and qualitative experiments validate the effectiveness and superiority of our plug-and-play design, underscoring the potential of leveraging generation discrepancies to model dense representations and encouraging further exploration of generative approaches for solving discriminative tasks.

## 1 Introduction

Recent breakthroughs in Diffusion Models (DMs) have empowered the field of visual generation for images (Rombach et al., 2022; Ruiz et al., 2023) and video (Cho et al., 2024; Ho et al., 2022), demonstrating their capacity of high-fidelity and diverse content synthesis. Meanwhile, there is a growing interest in unlocking DMs for performing the discriminative task of visual dense recognition (Xu et al., 2023a; Barsellotti et al., 2024a). However, similar to discriminative-model-based segmentation frameworks (Kirillov et al., 2023; Huynh et al., 2022; Zhou et al., 2022b), these DM-based methods rely heavily on large-scale pixel-level training datasets, which require costly and labor-intensive labeling efforts. To relieve this, this paper explores the potential of DMs in tackling the Inexact Segmentation (IS) problem, a more challenging task that achieves segmentation using only text or image-level class labels, essentially merging two existing settings: Text-Supervised Semantic Segmentation (TSSS) (Xu et al., 2022a; Ren et al., 2023; Xu et al., 2023b) and Weakly-Supervised Semantic Segmentation (WSSS) (Ahn & Kwak, 2018; Wang et al., 2020b).

One line of current DM-based IS research is dedicated to excavating and refining the image-text cross-attention map embedded in the noise predictor network (Wang et al., 2023b; Ma et al., 2023b). Specifically, these methods leveraged the object-shape-characterized self-attention module to refine the cross-attention map, yielding a segmentation mask for the query object. Another line of research focuses on treating a diffusion process as a self-supervised denoising task and employing a diffusion model as a general feature extractor (Xu et al., 2023a; Zhao et al., 2023). In these studies, diffusion models serve as attention-guiding feature extractors, *indirectly* assisting segmentation tasks. In contrast, research on using generative paradigms to *directly* optimize segmentation remains unexplored, leaving the fundamental generative ability of large-scale pretrained diffusion models underutilized.

In this paper, we delve into the generative nature of pretrained diffusion models to refine a coarse segmentation mask from inexact segmentation. Specially, we are inspired by cases that GPTs (Brown, 2020; Achiam et al., 2023) can generate responses closer to the alternative answers under certain prompts to solve discriminative tasks without any extra training. For visual diffusion models, better condition guidance similarly results in a smaller discrepancy between the generated and initial images. Under such an implication, we can use the discrepancy to obtain feedback to improve the condition itself. Prior work, DiffusionClassifier (Li et al., 2023a), has proved that using a correct text prompt leads to a better denoising result for a specific image, indicating better category classification. We incorporate this spirit into IS, a more challenging discriminative grounding task without pixel-level supervision. A new frame-

work, termed as **G4Seg**, is proposed, which leverages diffusion-based generation with coarse segmentation mask injection and the semantic discrepancy between the generated and initial image (as shown in Figure. 1). It is worth noting that G4Seg is an inference-only framework involving a large-scale pre-trained diffusion model without any extra training or fine-tuning.

Technically, to achieve refinement of the original mask in a generative manner, the image to be segmented should first be inverted into latent noise space or added with noise at a suitable time step. Then, the image is reconstructed with the condition, which includes the text prompt and the inexact mask. Under the imperfect mask, the generated image shows some discrepancy from the initial image. By means of the pixel-wise Hausdorff distance as a discrepancy metric, a semantic correspondence alignment methodology is designed for a better inexact segmented mask refinement.

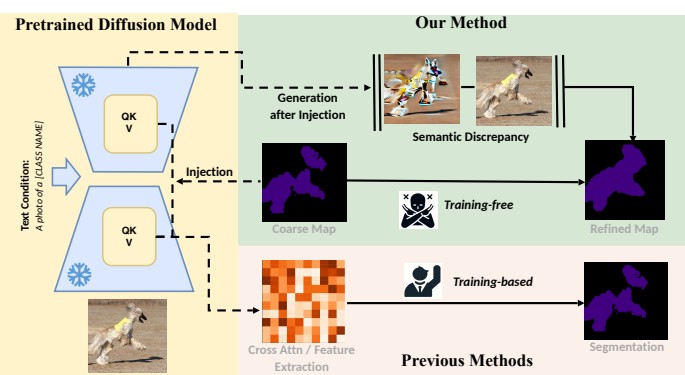

Figure 1: The comparison illustration. Previous DM-based methods mine out features or the cross-attention for generating the segmentation map with training. Our training-free method exploits the underlying semantic discrepancy after intervening the generation process to improve the segmentation map.

Our contributions can be summarized as follows:

- Different from the popular discriminative-based segmentation paradigm and previous DM-based training methods, we propose a novel training-free framework in a generation manner for inexact segmentation refinement empowered by the condition capacity of pretrained diffusion models.

- We are among the first attempts to leverage the discrepancy between original and generated images to refine the coarse mask by technically establishing a principled alignment to build the correspondence and updating the foreground probability of each pixel with its paired pixel.

- Our framework has achieved a consistent performance gain in both open-vocabulary and weakly supervised segmentation tasks on top of current state-of-the-art methods leveraging complementary knowledge sources from other post-refinement methods. The promising potential sheds light on using generative models to solve discriminative tasks without training.

## 2 RELATED WORK

**Diffusion Model-based Segmentation.** Diffusion Models (DMs), while demonstrating powerful image generation capabilities, have also exhibited emergent perception in object segmentation. A line of *segment-after-synthesize* works intuitively turns to Stable Diffusion (SD) (Rombach et al., 2022), representing the most powerful DM, to first synthesize extra high-quality pixel-level training datasets, which are then used to enhance the segmentor's performance (Li et al., 2023b; Nguyen et al., 2024; Ma et al., 2023a; Wu et al., 2023). Specifically, these two-stage methods either focus on exploiting the cross-attention map from SD for generating the first-stage mask or directly use the fused visual features from SD to train the second-stage segmentation module. Differentiating from such a two-stage pipeline, some methods shed light on directly transferring DMs into a discriminative segmentation model by generating the pixel-level output conditioned on the input image (Amit et al., 2021; Xu et al., 2023a; Burgert et al., 2022). For instance, ODISE (Xu et al., 2023a) proposed to train an SD-based segmentation framework by aligning the generated visual mask output with the corresponding caption and category labels. Contrary to these training-based frameworks, a stream of works Tang et al. (2022); Karazija et al. (2023); Barsellotti et al. (2024a;b); Marcos-Manchón et al. (2024); Yoshihashi et al. (2023), liberating from the costly pixel-level training process, has been dedicated to treating SD as an explicit training-free segmentor by directly mining its inner dense visual representation. OVDiff (Karazija et al., 2023) and FreeDA (Barsellotti et al., 2024b) tend to adopt the SD-based visual feature to generate the visual semantic prototype, serving as the nearest neighbor

| Related work | On-top-of | Training-free | GC | w/o DA | CAI |
|---|---|---|---|---|---|
| VPD (Zhao et al., 2023) | ✗ | ✗ | ✗ | ✗ | ✔ |
| ODISE (Xu et al., 2023a) | ✗ | ✗ | ✗ | ✗ | ✗ |
| OVDiff (Karazija et al., 2023) | ✗ | ✔ | ✗ | ✗ | ✗ |
| DiffSegmentor (Wang et al., 2023b) | ✗ | ✔ | ✗ | ✔ | ✔ |
| Freeda (Barsellotti et al., 2024c) | ✗ | ✔ | ✗ | ✗ | ✔ |
| DatasetDiffusion (Nguyen et al., 2024) | ✔ | ✔ | ✗ | ✔ | ✔ |
| UniGS (Qi et al., 2024) | ✗ | ✗ | ✔ | ✔ | ✗ |
| **G4Seg** (Ours) | ✔ | ✔ | ✔ | ✔ | ✔ |

Table 1: Comparison of related work across different criteria. 'GC' means Generative Content. 'DA' means Discriminative Assistance. 'CAI' means Cross Attention Initialization.

guiding the object segmentation in a zero-shot manner. DAAM (Tang et al., 2022), OVAM (Marcos-Manchón et al., 2024), Attn2mask (Yoshihashi et al., 2023), and DiffSegmenter (Wang et al., 2023b) explore and consolidate the usage of cross-attention across blocks, timestamps, and attention heads into a single attention map, which serves as a promising initial segmentation map. There is also one line of works unifying the generation and segmentation in one framework (Qi et al., 2024), which is trained end-to-end proposed for various segmentation and generation tasks.

Furthermore, we provide Table 1 to compare G4Seg and former methods in multiple aspects comprehensively, and more analysis can be found in Appendix K.

**Discriminative Models for Inexact Segmentation.** To liberate humans from exhaustive pixel-level annotation, recent years have witnessed extraordinary progress in Inexact Segmentation (IS), which aims to achieve a segmentation network equipped with coarse-grained labels. In this paper, we mainly discuss two derivative streams, i.e., Weakly-supervised Semantic Segmentation (WSSS) (Ahn & Kwak, 2018; Ahn et al., 2019; Zhang et al., 2021; Wang et al., 2020b; Zhu et al., 2023), and Text-Supervised Semantic Segmentation (TSSS) (Xu et al., 2022a; Zhang et al., 2023; Cha et al., 2022; Shin et al., 2022). WSSS regulates a segmenter trained with merely image-level labels. Most methods addressing WSSS focus on refining the seed areas generated by Class Activation Mapping (CAM) (Zhou et al., 2016), which merely captures the highly discriminative object regions. These methods, starting from early pooling-based mechanism modifications (Kwak et al., 2017) and regularized data augmentation enhancements (Zhang et al., 2021; Wang et al., 2020b), have gradually shifted to inter-pixel or semantic relation mining (Ahn et al., 2019; Xu et al., 2022b; Zhu et al., 2023).

TSSS aims to develop a segmentation model, trained with merely image-text pairs, that is able to segment arbitrary objects beyond predefined classes. This ability is also known as open-vocabulary segmentation. Most discriminative-model-based works addressing this can be categorized into two groups based on whether CLIP (Radford et al., 2021) is adopted for mask generation. The first category concentrates on extracting coarse localization features from CLIP through either the image-text cross-attention map (Shin et al., 2022; Cha et al., 2022; Zhou et al., 2022a) or the CAM (Zhou et al., 2016)-based attention map Lin et al. (2023), which are subsequently refined to achieve fine-grained segmentation performance. The second category, different from those CLIP-based training-free methods, focuses on enhancing plain Vision Transformers (ViT) (Dosovitskiy et al., 2020) by injecting grouping and clustering recognition from massive image-text training pairs, leading to a foundational segmentation model (Xu et al., 2022a; Ren et al., 2023; Luo et al., 2022; Zhang et al., 2023).

**Segmentation Post-Refinement** The segmentation post-refinement enhances the quality and precision of initial segmentation outputs by leveraging additional priors to address inaccuracies and improve overall performance. Dense CRF (Krähenbühl & Koltun, 2011) refines segmentation results by applying a fully connected Conditional Random Field (CRF) to the predicted probability map, leveraging pixel similarity and spatial relationships from the image. CascadePSP (Cheng et al., 2020) refines local boundaries with a novel refinement module, achieving pixel-accurate, class-agnostic segmentation across resolutions. SegRefiner (Wang et al., 2023c) enhances object masks using a discrete diffusion-based refinement approach. In comparison, our approach exploits image generation discrepancies empowered by a pretrained diffusion model to refine existing segmentation masks.

**Visual Correspondence.** Visual correspondence typically describes the matching relationship between specific points or features across different images that represent the same semantic, geometric,

or temporal meaning. Establishing semantic correspondence between different images can be crucially beneficial to various vision tasks, such as object segmentation (Liu et al., 2021; Zhang et al., 2020; Rubio et al., 2012; Xu et al., 2023c; Lan et al., 2021; Liu et al., 2023) and object recognition (Berg et al., 2005; Hao et al., 2013; Peng et al., 2017; Tang et al., 2020). For instance, Lan et al. (2021) utilized the semantic and geometric correspondence between images of the same region-of-interest features as consistency regularization for mask generation. Traditional correspondences have been modeled by hand-crafted features such as SIFT (Lowe, 2004) and SURF (Bay et al., 2006). With the rapid advances in deep neural architectures, a stream of works has intuitively developed a supervised training paradigm to find the correspondence (Lee et al., 2021; Zhao et al., 2021; Kim et al., 2017; Xiao et al., 2022). Nevertheless, these fully-supervised methods require massive correspondence annotations in the training datasets, limiting the model's scalability for practical applications. To address this issue, some works turn to correspondence models with only pose supervision (Wang et al., 2020a) or self-supervision (Wang et al., 2019; Jabri et al., 2020; Caron et al., 2021; Tumanyan et al., 2022). This work exploits DM-based semantic correspondence to improve the segmenter's performance explicitly.

## 3 METHOD

### 3.1 PROBLEM FORMULATION

Suppose we have an image $\mathcal{I}$ together with its coarse mask $\mathcal{S}_c$. It is worth noting that obtaining an inexact coarse segmentation mask $\mathcal{S}_c$ is simple and low-cost, achievable through methods like cross-attention extraction (Wang et al., 2023b) or by utilizing models such as CLIP (Lin et al., 2023). Towards our goal, we expect to use a pretrained diffusion model $\mathcal{M}$ to first obtain a generated image $\mathcal{I}_g = \mathcal{M}(\mathcal{I}_n; \mathcal{T}, \mathcal{S}_c)$, where $\mathcal{I}_n$ is the reversed embedding of $\mathcal{I}$ in the noise space and $\mathcal{T}$ is the text prompt. Then, by carefully comparing $\mathcal{I}$ and $\mathcal{I}_g$, we will get a mask with better quality $\mathcal{S}_r = \Phi(\mathcal{I}_g, \mathcal{I}, \mathcal{S}_c)$ using the algorithm $\Phi$.

### 3.2 PRELIMINARY

The visual diffusion model works by progressively adding noise to the image in the forward process and then using a deep network to recover the initial image from the pure noise in the backward process. In the forward process, the clean image $x_0$ is added with Gaussian noise scaled by a specific timestep $t$: $0 \le t \le T$, obtaining a noisy sample $x_t = \sqrt{\alpha_t}x_0 + \sqrt{1-\alpha_t}\epsilon$, where $\alpha_t$ and $\beta_t$ are the pre-defined noise schedules, $\alpha_t = \prod_{s=1}^{t}(1-\beta_s)$ and $\epsilon \sim \mathcal{N}(0, I)$. Then a deep learning network $\epsilon_\theta(x_t, t)$ is trained to predict the noise $\epsilon$ from $x_t$:

$$\mathbb{E}_{t \sim \mathcal{U}(0,T), \epsilon \sim \mathcal{N}(0,I)}[||\epsilon - \epsilon_\theta(x_t, t, y)||_2^2, \tag{1}$$

where $y$ is the condition. With a pre-trained diffusion model, a clean image can be generated from Gaussian noise $p(x_T) \sim \mathcal{N}(0, I)$ step by step by $x_{t-1} = \frac{1}{\sqrt{1-\beta_t}}(x_t - \frac{\beta_t}{\sqrt{1-\alpha_t}}\epsilon_\theta(x_t, t, y)) + \sigma_t z$, where $z \sim \mathcal{N}(0, I)$. This can be divided into two substeps. The first is to predict the original image $x_0$ (termed as $\tilde{x}_0$ to distinguish from $x_0$) using the current $x_t$ and the model prediction $\epsilon_\theta(x_t, t, y)$:

$$\tilde{x}_0 = f_\theta(x_t, t; y) = \frac{x_t - \sqrt{1-\alpha_t}\epsilon_\theta(x_t, t, y)}{\sqrt{\alpha_t}}. \tag{2}$$

Then, $x_{t-1}$ can be calculated as $x_{t-1} = \sqrt{\alpha_{t-1}}\tilde{x}_0 + \sqrt{1-\alpha_{t-1}-\sigma_t^2}\epsilon_\theta(x_t, t, y) + \sigma_t z$.

### 3.3 G4SEG: A MORE EFFICIENT GENERATIVE METHOD FOR INEXACT SEGMENTATION

In Section 3.1, we generate images conditioned on the coarse mask $S_c$. Normally, we can follow the method of full generation with null text inversion in (Mokady et al., 2023) to achieve near-perfect reconstruction for $\mathcal{I}$. To improve inference efficiency, we simplify the calculation of inverting $\mathcal{I}_n$ into a noise addition and denoising operation. For example, given a specific image $x_0$, we first select a candidate timestep $t_s$ and calculate the noisy sample $x_{t_s}$. Then, we shorten the whole generation process with only one step inference, using Eq. (2) to directly get the prediction $\tilde{x}_0$. As for the generation process intervened by the coarse prior $S_c$, we first transform the coarse mask $S_c$ into two masks respectively injected into the cross attention and self-attention of diffusion backbone, which is detailed in Section 3.3.1. Then, $\tilde{x}_0$ can be calculated under such a mask injection. Finally, the coarse mask is refined by employing the semantic correspondence alignment between $x_0$ and $\tilde{x}_0$ in Section 3.3.2. In the following, we concretely discuss the two critical components of our method.

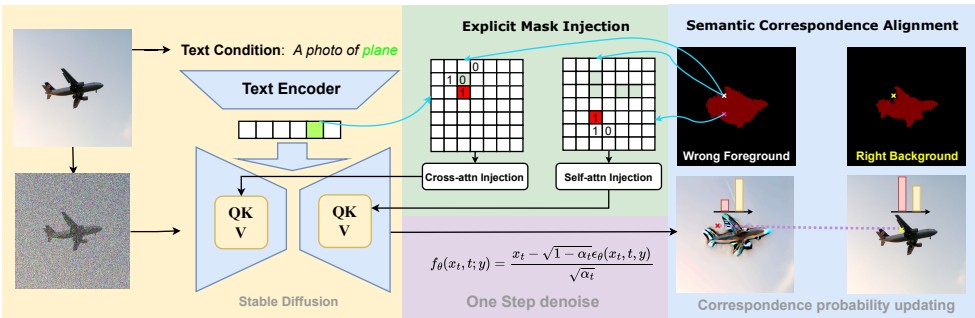

Figure 2: The overall framework of our proposed G4Seg. First, the noisy sample conditioned on the injected coarse mask is fed into the diffusion model to obtain the denoised image. Then, the foreground probability of each pixel is estimated with paired pixels in the semantic correspondence alignment. Finally, the updated segmentation mask is calculated from the pixel foreground probability.

### 3.3.1 EXPLICIT MASK INJECTION

In Stable Diffusion, the textual prompt $\mathcal{T}$ is first tokenized and fed into the CLIP text encoder, forming a textual embedding. The denoising U-Net then utilizes cross-attention mechanisms with the embedding to leverage textual information for conditioning. At the same time, self-attention is employed to model the relationships between pixels, which can be leveraged for better generation. In this study, we utilize the aforementioned features of cross-attention and self-attention in our diffusion model to inject our prior coarse segmentation mask into the inference process.

Specifically, in the attention layers of diffusion models, the intermediate image feature is first mapped as a query and updated via calculating the attention maps $A \in \mathbb{R}^{q \times k}$ with $A = \text{softmax}\left(\frac{QK^\top}{\sqrt{d}}\right)$, where $q$ and $k$ are the lengths of the query $Q$ and the key $K$, which are derived from the context. The context could either be a text embedding or the image feature itself, noted as cross-attention and self-attention, respectively. We incorporate the coarse mask as a representation of the ideal correlations between pixels and textual embeddings, integrating it into the generation process of our diffusion model, following the spirit of DenseDiffusion (Kim et al., 2023).

For clarity, we flatten the 2D image mask into a 1D signal, facilitating alignment with the 1D textual signal. We assign a superscript to such signals for representation, $e.g.$, $S_c^{1D}$ denotes the coarse mask that is flattened into 1D. For a coarse mask provided for the category $c$ with the name $T_c$, we prepare the prompt $\mathcal{T}$ as "A photo of $T_c$" and map it to a textual embedding as the key feature. Suppose the index set of $T_c$ in the textual embedding is $\alpha(T_c)$ [1]. The injection mask for cross attention $\mathcal{A}_{\text{cross}} \in \mathbb{R}^{q \times k}$ is designed as:

$$\mathcal{A}_{\text{cross}}(i, j; S_c) = \begin{cases} 1 & \text{if } j \in \alpha(T_c) \text{ and } S_c[i] = 1 \\ 0 & \text{otherwise} \end{cases}. \tag{3}$$

$\mathcal{A}_{\text{cross}}(i, j; S_c)$ represents the relation between two types of signals, image, and text, which is set to 1 once the textual embedding and the foreground image token is matched, otherwise 0. Similarly, we can define the injection mask for self-attention $\mathcal{A}_{\text{self}} \in \mathbb{R}^{q \times q}$. However, as the self-attention performs between image tokens, we can compute the mask on the internal $S_c$, which is formulated as follows

$$\mathcal{A}_{\text{self}}(i, j; S_c) = \begin{cases} 1 & \text{if } S_c[i] = S_c[j] = 1 \\ 0 & \text{otherwise} \end{cases}. \tag{4}$$

Given two injection masks $\mathcal{A}_{\text{self}}$ and $\mathcal{A}_{\text{cross}}$, we can respectively intervene in the computation of the cross attention and the self-attention in image generation with the pretrained diffusion model

$$A'_{\text{cross}} = \text{softmax}(\frac{QK^\top + \alpha\mathcal{A}_{\text{cross}}}{\sqrt{d}}), \quad A'_{\text{self}} = \text{softmax}(\frac{QK^\top + \alpha\mathcal{A}_{\text{self}}}{\sqrt{d}}), \tag{5}$$

---

[1] The index of the class name is represented as a set, as a single long word may correspond to multiple token embeddings, or the class name may consist of two or more words.

where the $\alpha$ is the injection weight. By incorporating the intervention of the coarse mask $S_c$ by Eq. 3, Eq. 4 and Eq. 5, the image generation result $\tilde{x}_0$ is affected. In the next section, we will illustrate how to employ the gap between the reconstructed image $\tilde{x}_0$ and the original image $x_0$.

### 3.3.2 SEMANTIC CORRESPONDENCE ALIGNMENT

With explicit mask injection, we have obtained a model that can generate images conditioned on the coarse mask. In other words, without loss of generality, we have a generative model $p(x|S)$, where $S$ is a given mask and $x$ is the target image. However, in a segmentation task, we actually want to find $\max_S p(S|x)$ given an image $x$. Intrinsically, they can be connected by using Bayes' Law as below

$$\max_S p(S|x) = \max_S \frac{p(x|S)p(S)}{p(x)} \Leftrightarrow \max_S p(x|S), \tag{6}$$

since $p(x)$ and $p(S)$ should be constant for a specific $x$. Here, the segmentation task can be treated as a conditional generation problem, which fits our intuition that *with more accurate mask condition, the probability of generating $x$ is more likely to be maximized.* Following this spirit, we assume that $p(x|S)$ follows a distribution that is inversely related to $d(x, \tilde{x}(S))$, namely $p(x|S) \propto -d(x, \tilde{x}(S))$, where $x(S)$ denotes the corresponding generation conditioned on the mask $S$, and $d$ represents an image-wise distance measure. Consequently, the problem reduces to $\min_S d(x, \tilde{x}(S))$. In this study, we realize the image-level distance measure by means of the pixel-level Hausdorff distance (Huttenlocher et al., 1993), denoted as $d_{\text{Haus}}(\cdot, \cdot)$, which provides us the inspiration of transforming the optimization into a semantic correspondence alignment based on image discrepancy, formulated as below.

$$\max_S p(x|S) \xrightarrow{\text{reduce}} \min_S d_{\text{Haus}}(x, \tilde{x}(S)) : S[j] \leftarrow S[j] + \gamma \frac{\partial D(x[\delta_j], \tilde{x}(S)[j])}{\partial S[j]}, \tag{7}$$

where $\tilde{x}(S)[j]$ denotes the $j$th pixel in the generated image $\tilde{x}(S)$ and $\delta_j$ denotes the index of the corresponding pixel in the original image $x$ that requires to be searched. $D$ is a pixel metric based on the semantic gap between two pixels. The detailed deduction can be found in Appendix C. For a specific category as foreground, if we treat $S[j]$ as the foreground probability, we can interestingly observe that $S[j]$ is updated based on the discrepancy ($D$ in the Equation, which denotes as a semantic gap) between the probability of the pixel in the original and generated images.

Despite the potential insight inherent in Eq. 7, it is intractable due to the discrete operation implemented on the coarse mask $S$ in Eq. 3 and Eq. 4. However, we can follow its spirit to build a semantic correspondence alignment to achieve a similar goal. That is: 1) we first find the optimal pixel alignment $\delta_j$ as in Eq. 7; 2) and then we use a simple linear mixing between paired pixels in the generated and initial images to approximate the segmentation (foreground) mask updating direction. For the first step, we use a predefined feature extractor $F(\cdot)$ to embed the generated and original images are into the feature space, denoted as $F(\tilde{x}(S))$ and $F(x)$. For the $j$th pixel in the generated image, the corresponding point $\delta_j$ can be searched via:

$$\delta_j = \arg\min_{j'} \mathcal{D}(F(x)[j'], F(\tilde{x}(S))[j]), \tag{8}$$

where $\mathcal{D}(\cdot, \cdot)$ denotes the cosine similarity metric defined in the feature space for semantic correspondence alignment. For a specific pixel, we obtain the pixel-wise feature from the image feature and then search for the pixel in the original image that has the smallest distance. Then, for the second step, we can estimate the foreground probability $S^\star$ at position $j$ as follows,

$$S^\star[j] = \beta S[j] + (1 - \beta)S[\delta_j], \tag{9}$$

where $\beta$ is the mixing coefficient. Finally, with the refined foreground probability $S^\star[\cdot]$ of each pixel, we obtain the refined segmentation mask.

Intuitively, the linear mixing of paired pixels could adaptively refine the pixels in the wrongly segmented area. For instance, if the foreground area is wrongly segmented as background (under-segmented, often near the edge), under the condition of a mask that does not fully cover the foreground, the entire foreground object tends to shrink inward after the generation. The pixel in this under-segmented area can be paired with a shrunken foreground area in the generated image, which is in the interior of the object with higher foreground probability. Then after probability mixing, the pixel is more likely to be classified as foreground. The analysis remains similar to the over-segmented area. More analysis and examples can be found in Appendix. F.

# 4 EXPERIMENTS

## 4.1 IMPLEMENTATION DETAILS

**Datasets and Evaluation Metric.** Following Lin et al. (2023); Xu et al. (2022a); Zhang et al. (2023), we evaluate G4Seg on three benchmarks, i.e., PASCAL VOC12 (20 foreground classes) Everingham et al. (2015), PASCAL Context (59 foreground classes) Mottaghi et al. (2014), and MS COCO Object 2014 datasets Lin et al. (2014) (80 foreground classes). All of these datasets contain 1 extra background class. During the inference, only the image-level (class) label is used to generate the mask. The mean Intersection-over-Union (mIoU) is adopted as the evaluation metric (%).

**Inference Settings** Our model is fully based on Stable Diffusion 2-1Rombach et al. (2022), which is trained on LAION Schuhmann et al. (2022). In our experiment, all images are resized to (512, 512). All experiments are merely conducted on 1 RTX 3090 GPU equipped with 24 GB of memory without any extra training. Our method, working in an on-top-of manner, follows a *refine-after-generate* paradigm: generating the masks from the selected mask-free baseline first and then refining them via our proposed method without additional training. For the mask generation process, we strictly follow the settings in the selected baselines. For explicit mask injection, our parameter follows the DenseDiffusion Kim et al. (2023) and the added noise step is 400. For the semantic correspondence alignment, the feature extractor we adopt is a CLIP image encoder to better distinguish between generated and initial image. The pixel correspondence mixing coefficient $\beta$ is set to 0.8 for open-vocabulary segmentation and 0.9 for weakly-supervised semantic segmentation. For each specific class, We treat all other segments as background and update the current segment logit independently. Then the final segmentation is refined with updated logit after normalization. As the framework is dedicated to a mask refining task, we only select the confusion areas in the coarse mask $S_c$, providing the upper and lower bounds of the foreground probability. The confusion area is selected as where the foreground probability value is within the range of [0.2, 0.6] of the maximum foreground probability value for the current class, and the distance to the edge does not exceed 40 pixels.

## 4.2 INEXACT SEGMENTATION PERFORMANCE

**Performance on TSSS.** Here we first evaluate the performance of our method in TSSS. Table 2 lists the mIoU of 11 state-of-the-art (SOTA) methods on the validation of PASCAL VOC12, PASCAl Context, and COCO Object. Notably, these methods are categorized into two splits, i.e., *training-based* and *training-free*, and we implement our on-top-of method based on 3 methods (1 training-based + 2 training-free).

Note that our method does not involve any training process. As shown in this table, it is clear that our method, regardless of the training paradigm, could achieve an overall improvement compared to all the adopted baseline methods, with an average ela-

Table 2: Comparison with TSSS methods.

| Methods | VOC12 | Context | COCO |
|---|---|---|---|
| *Training-based* | | | |
| ViL-Seg (Liu et al., 2022) | 34.4 | 16.3 | 16.4 |
| TCL (Cha et al., 2022) | 51.2 | 24.3 | 30.4 |
| GroupViT (Xu et al., 2022a) | 52.3 | 22.4 | 20.9 |
| ViewCo (Ren et al., 2023) | 52.4 | 23.0 | 23.5 |
| SegCLIP (Luo et al., 2022) | 52.6 | 24.7 | 26.5 |
| PGSeg (Zhang et al., 2023) | 53.2 | 23.8 | 28.7 |
| OVSegmentor (Xu et al., 2023b) | 53.8 | 20.4 | 25.1 |
| G4Seg+*GroupViT* | 53.4+1.1 | 23.9+1.5 | 22.1+1.2 |
| | | | |
| *Training-free* | | | |
| ReCo (Shin et al., 2022) | 25.1 | 19.9 | 15.7 |
| MaskCLIP (Zhou et al., 2022a) | 38.8 | 23.6 | 20.1 |
| SCLIP (Wang et al., 2023a) | 59.1 | 30.4 | 30.5 |
| DiffSegmenter (Wang et al., 2023b) | 60.1 | 27.5 | 37.9 |
| G4Seg+*SCLIP* | 59.8+0.7 | **31.3**+0.9 | 30.9+0.4 |
| G4Seg+*DiffSegmenter* | **60.6**+0.5 | 28.1+0.6 | **38.5**+0.6 |

tion of **0.77**%, **1.00**%, and **0.73**% across these three benchmarks. Additionally, with such prominent improvement, our method yields new SOTA performance against all methods in TSSS. Figure 3 shows some illustrative samples for a visualized comparison, validating the effectiveness and superiority of our method in open-domain segmentation refinement.
**Performance on WSSS.** Here we compare our methods with a line of works in WSSS. As downstream training is required, WSSS evaluates the model's ability to segment task-specific objects. In this way, to evaluate the effectiveness of our method in task-specific learning, Table 3 reports the performance of our method in comparison with 8 prevailing WSSS frameworks. Here we would like to emphasize that two post-processing refining mechanisms are commonly utilized in WSSS, i.e., RW (Ahn et al., 2019) and dCRF (Chen et al., 2017), which helps *refine the coarse Seed into the fine-grained Mask*.

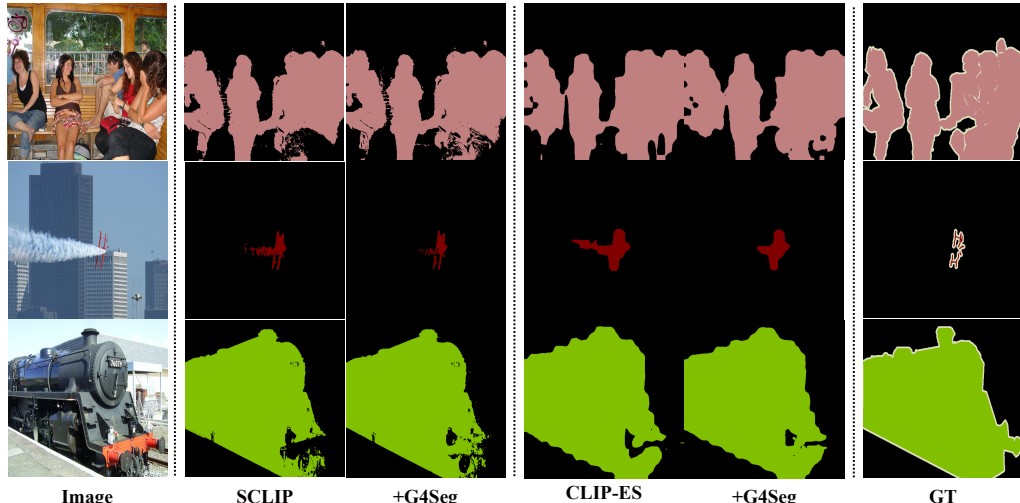

| Image | SCLIP | +G4Seg | CLIP-ES | +G4Seg | GT |

Figure 3: Qualitative results on PASCAL VOC12. Compared with the baseline, G4Seg could further segment the object in a more complete and delicate way.

As shown in Table 3, our method based on two WSSS frameworks achieves an overall consistent improvement compared to the adopted baselines, leading to an average accuracy increase of **2.0%** (**1.2%**) on Seed (Mask). We observed the average improvement brought by G4Seg on samples with varying initial mask quality: for samples with an initial IoU below 40, G4Seg achieved an improvement of 0.2; for those with an initial IoU between 40 and 80, it provided a significant boost of 1.9; for samples with an initial IoU between 80 and 100, it enhanced performance by 1.1.

These experimental results further validate the versatility of our method in domain-specific segmentation. In this way, our approach with CLIP-ES yields a new SOTA performance in WSSS, further demonstrating the excellence of our training-free method in zero-shot IS. Figure 3 showcases some illustrative samples that are produced from the adopted baseline and our methods. It is observed that our method could process fine-grained segmentation by refining the object boundary. More results are provided in Appendix O.

Table 3: Comparison with WSSS methods on VOC12 *train*. The mask is generated from Seed refined with Post-processing (Post.) approaches. * denotes that Zhu et al. (2023) adopts a designed self-training strategy. All these methods merely adopt the image-level labels during the inference.

| Methods | Post. | Seed | Mask |
|---|---|---|---|
| CAM Ahn & Kwak (2018) | dCRF | 48.0 | 52.4 |
| IRN Ahn et al. (2019) | RW+dCRF | 48.5 | 63.5 |
| SEAM Wang et al. (2020b) | RW+dCRF | 55.4 | 63.6 |
| MCTformer Xu et al. (2022b) | RW+dCRF | 61.7 | 69.1 |
| ViT-PCM Rossetti et al. (2022) | dCRF | 67.7 | 71.4 |
| ToCo Ru et al. (2023) | - | 73.6 | 73.6 |
| WeakTr Zhu et al. (2023) | Self-Training* | 66.2 | 76.5 |
| CLIP-ES Lin et al. (2023) | dCRF | 70.8 | 74.9 |
| G4Seg+*CAM* | dCRF | 50.8+2.8 | 54.2+1.8 |
| G4Seg+*CLIP-ES* | dCRF | **72.0**+1.2 | **75.4**+0.5 |

### 4.3 ABLATION STUDIES

In this Section, unless specifically specified, we use the Seed of G4Seg with CLIP-ES to implement all ablation studies on PASCAL VOC12 in detail, which mainly contains the effectiveness of the modules in G4Seg, the influence of time step, and some illustrative visualized results.

**Effectiveness of Individual Module.** Table 4 presents the effectiveness of each individual module in G4Seg. As shown in this table, adding EMI could explicitly bring a certain elation (**+0.5%**) compared with the baseline, indicating the benefits of mask injection during the denoising stage. Additionally, further improvements

Table 4: Ablation studies on the modules in G4Seg.

| Baseline | EMI | SCA | CF-[0.2,0.6] | CF-[0.1,0.7] | mIoU (%) |
|---|---|---|---|---|---|
| ✔ | | | | | 70.8 |
| ✔ | ✔ | | | | 71.3+0.5 |
| ✔ | ✔ | ✔ | | | 71.7+0.9 |
| ✔ | ✔ | ✔ | ✔ | | **72.0**+1.2 |
| ✔ | ✔ | ✔ | | ✔ | 71.6+0.8 |

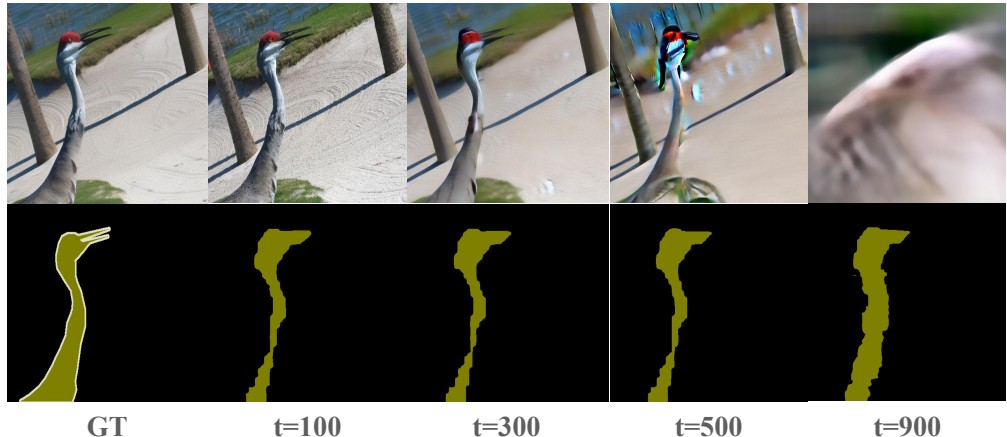

Figure 4: Visualized analysis of G4Seg under different denoising timesteps.

achieved through SCA (**+0.9%**) demonstrate that establishing a correspondence between the mask-injected image and the original image can emphasize the importance of key matching points for fine-grained segmentation. We also propose the CF strategy to further improve the performance of SCA by matching and modifying the most uncertain points. Consequently, it is observed a proper setting of the filtering range could yield the boosting of G4Seg (**+1.2%**), achieving a final **72.0%** performance together with all modules.

**Different Timesteps.** G4Seg adopts the fixed noising-denoising step for the generated image. To investigate the impact of the denoising timestep, we conduct our method by setting different timesteps obtained from {100, 200, 300, 400, 500}. As shown in Figure 5, it can be observed that our method is overall robust to the timestep due to a merely small performance fluctuation. The best performance is achieved at step 400, and then the larger/smaller timestep could yield a performance decrease. Figure 4 shows one illustrative sample generated with different timesteps. Clearly, a timestep that is too small, representing a minor perturbation to the original image, would reasonably yield insufficient knowledge injection. Conversely, a timestep that is too large results in a substantial visual discrepancy between the generated and original images, leading to invalid correspondence matching.

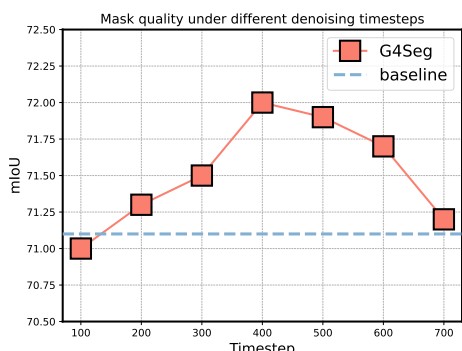

Figure 5: The mask quality under different noise scale with correspondent timestep.

**Involvement of Null-text Inversion.** G4Seg utilizes the difference between the generated and original image to help refine the mask. Due to the single-step noising-denoising process, it is hard to flawlessly reconstruct the original image. Intuitively, here we explore *whether better reconstruction could bring more explicit improvement.* To this end, we introduce Null-text Inversion (NtI) for our method, achieving near-perfect reconstruction by finding the corresponding initial noise during the inversion. Table 5 reports the performance and the inference speed comparison between our method and NtI-involved paradigm. Interestingly, the involvement of NtI simply showcases the marginal improvement as expected. Figure 6 presents some visualized samples. Despite better image reconstruction, there is a low discrepancy in the segmentation performance between NtI-free and -based methods.

Table 5: The influence of Null-text Inversion (NtI). The unit of speed is second(s) for processing one sample.

| Method | G4Seg+GroupViT / + NtI | G4Seg+CLIP-ES / + NtI |
|---|---|---|
| mIoU | 54.0 / 54.2 | 72.0 / 72.1 |
| Speed(s) | +1.2/5.5 | +1.1/5.2 |

**Computational Analysis.** Our method is training-free; it directly uses a pre-trained diffusion model, thus saving a significant amount of resources that would otherwise be consumed during training.

Unlike the generative process of diffusion, which requires multiple forward passes, our method only requires a single forward pass for a specific segment. But there are still some limitations if there are segments in a single image, the model should forward multiple times. However, with the development of the composed diffusion process Wu et al. (2024) where multiple object priors can be injected in a single forward pass, the computational cost of our method will be significantly reduced. The correspondence calculation is performed on a much more compact space with lower dimensions and other restrictions declared in Section. 4.1, resulting in a significant saving in computational cost.

**Comparison with other mask refinement methods** In this section, we compare with three other mask refinement methods: CascadePSP (Cheng et al., 2020), SegRefiner (Wang et al., 2023c), and Dense CRF (Perez & Wang, 2017). CascadePSP and SegRefiner focus on improving segmentation and require pixel-wise annotations for training. As for semantic segmentation, these methods may focus more on improving segmentation around the boundary. Dense CRF is a widely used traditional method that leverages priors constructed from the image itself to refine the coarse mask.

According to the Table 6, we make the following comments: Although CascadePSP and SegRefiner use many pixel-level labels for training, the performance improvement in in-exact semantic segmentation is still quite limited. DenseCRF, as a method that refines coarse predictions by leveraging the information of image formation, improves the

Table 6: Comparison with other mask refinement methods

| Methods | VOC | Context |
|---|---|---|
| SCLIP | 59.1 | 30.4 |
| +G4Seg | 59.8(+0.7) | 31.3(+0.9) |
| +SegRefiner (Wang et al., 2023c) | 59.3(+0.2) | 30.7(+0.3) |
| +CascadePSP (Cheng et al., 2020) | 59.5(+0.4) | 30.9(+0.5) |
| + CRF (Krähenbühl & Koltun, 2011) | 60.9(+1.8) | 31.2(+0.8) |
| +G4Seg + CascadePSP | 60.1(+1.0) | 31.6(+1.2) |
| +G4Seg+Dense CRF | **62.1(+3.0)** | **32.0(+1.6)** |

initial segmentation with a significant margin when the number of classes is limited. However, as the number of classes increases, the improvements achieved by DenseCRF become less significant. The improvements of our method are roughly comparable to those of the DenseCRF on Context dataset. Since the source of segmentation knowledge in our method differs from that of other approaches (CascadePSP relies on annotations, and DenseCRF leverages image-based priors), G4Seg can be further combined with these methods to achieve additional improvements.

**General Applications on other forms of Inexact Segmentation.** We further investigated the impact of G4Seg on improving IS under weak label forms such as box and scribble, where the complete coarse mask is firstly obtained with these labels, the results are shown in Table 7. The experi-

Table 7: The segmentation results with other inexact forms of weak labels with boxes, points and scribbles.

| IS form | Point | Box | Scribble |
|---|---|---|---|
| SPML (Ke et al., 2021) | 72.7 | 75.3 | 72.5 |
| SPML+G4Seg | +1.5 | +1.1 | +1.6 |

ments have demonstrated that our method provides consistent improvements across various forms of inexact segmentation under weakly supervised labels.

## 5 CONCLUSION

This paper explored an intuitive yet feasible training-free solution based on Stable Diffusion (SD), a representative large-scale text-to-image diffusion model, to tackle the challenging vision task of Inexact Segmentation (IS), which aims at achieving segmentation using merely texts or image-level labels as minimalist supervision. Most SD-based trials, following the discriminative-model-exploited pipelines, fall into the pure exploitation of the visual dense representations inherently arising from the inner attention mechanism. In contrast, this paper emphasized the underlying generation prior in SD, i.e., the pattern discrepancy between the original and mask-conditioning reconstructed images, to encourage a coarse-to-fine segmentation refinement by progressively aligning the generated-original representations. Furthermore, we proposed establishing the pixel-level semantic correspondence between the generated-original patterns, yielding a delicate correction towards flawless segmentation for the matched point. Through quantitative and qualitative experiments, we have demonstrated the effectiveness and superiority of this plug-and-play design. Our results highlight the potential of utilizing generation discrepancies to model dense representations in diffusion models. We hope this work inspires further exploration of diffusion models in discriminative tasks.

ETHICS STATEMENT

Note that our method uses the diffusion model to generate the image data, which may raise ethical and moral concerns. Specifically, these generated data could defame individuals and spread misinformation, posing serious threats to personal reputations and societal trust. Besides, there is the risk of generating inappropriate or harmful content, which can have psychological and social repercussions. The online-collected benchmark used in our paper contains a wide range of objects, and the generated artificial images based on these benchmarks may have a biased understanding for humans and deep networks to learn the visual patterns. Furthermore, our method aims to generate a dense representation from simply human-annotated text supervision, which may also lead to biased orientation if the annotation lacks certain regulations.

REPRODUCIBILITY STATEMENT

In order to ensure the reproducibility of our work, we will provide access to the full implementation of our methods, including all necessary code and scripts, upon acceptance. An anonymous link to our code repository will be shared during the discussion phase of the review process. This repository will contain detailed instructions for reproducing the experiments, including dataset preparation, model training, and evaluation procedures. Additionally, the exact configurations (e.g., hyperparameters) are illustrated in Section. 4.1 used to generate the reported results to facilitate easy replication.

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

## A  LIMITATIONS AND FUTURE WORK

Though promising performance is achieved by our method, such a DM-based method inevitably meets the comparably slow-mask-inference issue due to the sampling denoising process. Besides, since SD is simply trained on natural images, our training-free method may not be applicable in some non-natural image domains, such as medical and agricultural imaging. Finally, due to resource limitations, we do not implement the latest SD version (SD-XL, SD3, Flux), and additional tuning on our method is also not achieved, both of which shall lead to better segmentation performance.

## B  TIME AND MEMORY EFFICIENCY

Our G4Seg could be implemented on simply 1 RTX 3090 GPU, generating 1 mask at a time and occupying 15GB. Since our method simply requires a direct single-step noising-denoising process to the original image, G4Seg could finish the inference of all 1449 images in VOC12 validation images within 1.5 hours (3 seconds per image), leading to a reasonable level of computational efficiency. Note that adopting multiple GPUs or multiprocessing could further speed up the inference process. In fact, we implement our G4Seg with 4 3090 GPUs. In this way, the inference time is reduced to about 18 minutes.

## C  A HAUSDORFF DISTANCE VIEW OF CORRESPONDENCE ALIGNMENT

Here we illustrate our method in a more theoretical view. Suppose we have a mask $S$ conditioned generation model, which could estimate $p(x|S)$. Then, we want to inverse this process with $p(S|x)$ which denotes that given a $x$ the $S$ distribution should be estimated. So in a segmentation task, we want to estimate:

$$\max_S p(S|x), \tag{10}$$

where $x$ denotes specific samples. Owing to the law of condition probability:

$$p(S|x) = \frac{p(x|S)p(S)}{p(x)}.$$

For the given $x$ and suppose all the segmentation masks share the same probability, we omit the $p(x)$ and $p(S)$ terms. Then the final result is equivalent as the:

$$\max_S p(x|S) \quad \text{with specific } x. \tag{11}$$

where indicates our institution, **with accurate mask condition, the probability of generating $x$ is maximized.** This is truly our basic stone.

Here we make further assumption, owing to the Gaussian essence of diffusion generation, the $p(x|S)$ could be estimated by:

$$p(x|S) \propto \exp(-d(x, \tilde{x}(S))^2), \tag{12}$$

where the $\tilde{x}(S)$ denotes the generating $\tilde{x}$ based on S. Then the problem is equivalent to $\min_S d(x, \tilde{x}(S))$. The problem becomes, finding a more appropriate mask, then minimum the gap between the mask-conditioned generation and initial image.

Then we based on this update the $S$ with stochastic gradient descent:

$$S = S + \gamma \frac{\partial d(x, \tilde{x}(S))}{\partial S}, \tag{13}$$

where $\gamma$ denotes the step size. Then we consider a Hausdorff distance between two images $(A, B)$ with pixel-wise $(a, b)$ distance:

$$H(A, B) = \sup_{a \in A} \inf_{b \in B} D(a, b), \tag{14}$$

where $D$ denotes the pixel-wise distance to distinguish from the image-wise distance $d$. Here we consider the initial image and conditioned generated image,

$$H(\tilde{x}(S), x) = \sup_{\tilde{x}(S)_j \in \tilde{x}(S)} \inf_{x[i] \in x} D(x[i], \tilde{x}(S)[j]), \tag{15}$$

where the $[i, j]$ indicates the i'th and j'th pixel of the initial and generated image. If we carefully look at the $\inf_{x[i] \in x} D(x[i], \tilde{x}(S)[j])$, **the term indicates the correspondence pixel among all $x[i]$s in $x$**

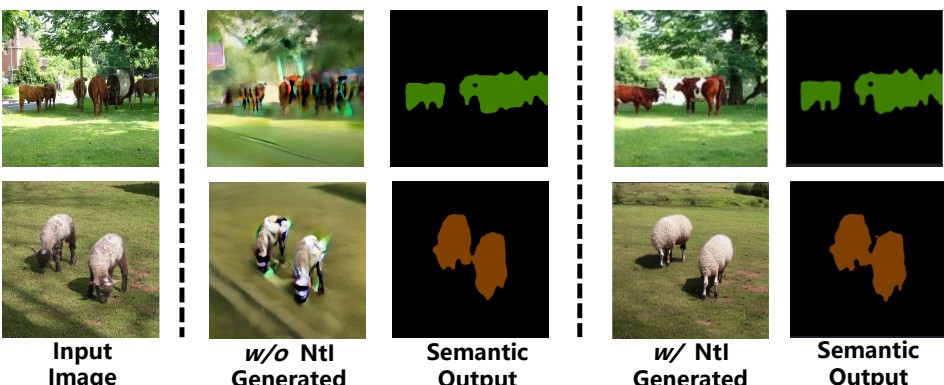

Figure 6: Visualized comparison between G4Seg *w/* NtI and *w/o* NtI.

**with minimum distance towards** $\tilde{x}(S)[j]$. Here we consider an equivalent formation substituting superb with summation.

$$H'(\tilde{x}(S), x) = \sum_{\tilde{x}(S)[j] \in \tilde{x}(S)} D(x[\delta_j], \tilde{x}(S)[j]), \tag{16}$$

where $x_{\delta_j}$ denotes the correspondence point with $\tilde{x}(S)[j]$. With specific $j$'th pixel $\tilde{x}(S)[j]$, we substitute Eq. 16 into Eq. 13, then we obtaining:

$$S[j] \leftarrow S[j] + \gamma \frac{\partial D(x[\delta_j], \tilde{x}(S)[j])}{\partial S[j]}, \tag{17}$$

where this could be treated as mask optimization and updating process.

## D    VISUALIZATION FOR G4SEG AND NULL TEXT INVERSION

The results could be found in Figure. 6.

## E    CORRESPONDENCE ANALYSIS.

Our proposed SCA explicitly refines the mask by building the feature-level semantic correspondence. Figure 7 presents the visualized correspondence matching map. SCA builds a one-to-one mapping between the original (stars) and mask-injected generated images (circles). The matched pixel from the generated image (marked by the small circles) reflects the same semantic content as the original image (marked by the stars). However, with the coarse mask injection, the generated image shall have wrongly-recognized regions for the query object. Specifically, there is a generated semantic of "train" for the railroad in the generated image, which is the result of the over-segmented coarse mask (marked by the green box). With the help of correspondence alignment, the mis-segmented pixel is corrected to embrace the appropriate object regions, relieving the over-segmented regions. In this way, we observe that the incorporation of correspondence helps improve the boundary regions. Such fine-grained refinement further validates the effectiveness of G4Seg, demonstrating the rationality of adopting generation discrepancy in segmentation which is consistent with the discussion in Section 3.3.2.

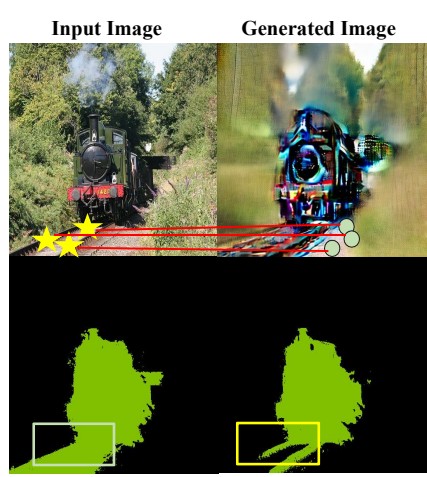

Figure 7: Visualization analysis on SCA. SCA is able to correct the wrongly segmented pixels based on the generated-original image discrepancy.

# F ADAPTIVE ADJUSTMENT
## FOR OVER/UNDER-SEGMENTED AREA

Different types of errors can introduce various impacts on the generated results. We make a more detailed discussion on the Figure 8. We divided these errors into two categories and discussed the points in each area:

## F.1 OVER-SEGMENTED

**Definition**: segmenting some background as foreground.

**Phenomenon**: The generated area tends to expand, incorporating the semantic of the object into areas that were originally background, as shown in the first row of Figure 9.

**Segmentation refining**: The corresponding point moves to the exterior with a lower probability. Then the mixing in Eq. 9 would lead to a lower foreground probability, the point is more likely to be recognized as background correctly.

## F.2 UNDER-SEGMENTED

**Definition**: segmenting some foreground as background.

**Phenomenon**: The generated object tends to shrink, converting areas originally belonging to the object into the background, as shown in the second row of Figure 9.

**Segmentation refining**: The corresponding point moves to the interior of the object with increasing foreground probability. Then after the points probability mixing, the point is more likely classified as foreground correctly.

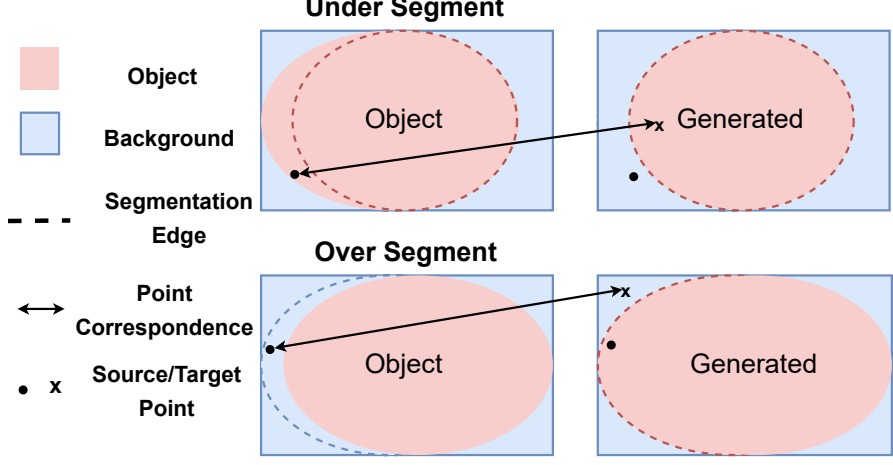

Figure 8: the linear mixing of foreground probability of paired pixels could adaptively adjust the over-/under segmented area.

In summary, our method generates corresponding defective images based on the flaws in the existing segmented mask. The mixed probability is then adaptively adjusted according to different scenarios. A visualization result can be found in Figure 9.

# G MASK INJECTION BOTTLENECK

In Sect. 3.3.1, we introduced a mask conditioning method, which is based on cross-attention and self-attention. These attention-based generation methods do not perform well in a mask-conditioned generation. If we adopt a stronger mask conditioning method, such as ControlNet, the performance would significantly improve, as shown in the following Table:

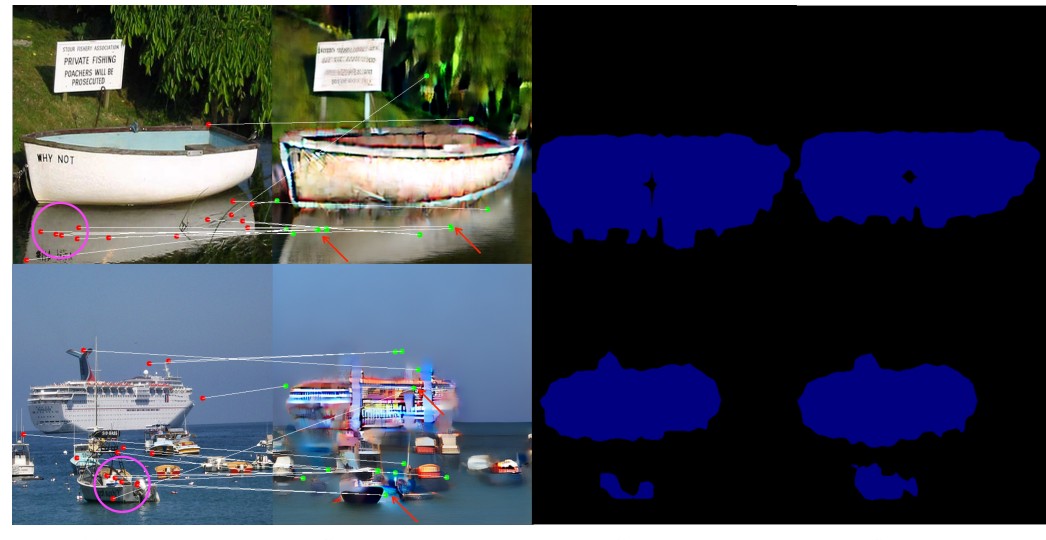

| Initial Image | Generated Image | Coarse Mask | Refined Mask |

Figure 9: Visualization of correspondence and segmentation refining. Random 15 points are selected for visualization. In the ships in the second row, the coarse segmentation reveals that the middle part of the smaller ship is missing. The semantics of pixels in the middle part are eroded by the background in generated image. The points under-segmented in red circles are mapped to the edges of the ship and the hull of the larger ship with higher target probability.

| Method | CLIP-ES VOC | SCLIP COCO | SCLIP Context |
|---|---|---|---|
| G4Seg+EMI(Attn Injection) | 72.0 | 30.9 | 31.3 |
| G4Seg+EMI(Controlnet (Zhang et al.)) | 74.1 | 33.1 | 33.8 |

## H  ON TOP OF FULLY/SEMI-SUPERVISED METHODS

**Fully/semi-supervised open-vocabulary semantic segmentation** To better evaluate our methods, we build G4Seg on top of some fully/semi-supervised open-vocabulary semantic segmentation methods:

- OVAM (Marcos-Manchón et al., 2024): OVAM uses manually annotated masks of generated images to update token embeddings, which are then used to generate more images and corrected cross-attention-based pseudo masks.

- DeOP (Han et al., 2023) DeOP is inherently a fully supervised method trained with precisely annotated pixel labels.

| Methods | VOC | Context |
|---|---|---|
| OVAM | 61.2 | 28.3 |
| +G4Seg | 62.1(+0.9) | 28.9(+0.6) |
| DeOP | 91.7 | 48.8 |
| +G4Seg | 92.1 (+0.4) | 49.3(+0.5) |

**Fully supervised closed setting** Our method relies on a pre-trained diffusion model and allows for sample-wise segmentation improvement by providing the image and its corresponding coarse mask. For closed-set semantic segmentation we conduct our method on ADE20k with three fully supervised segmentation approaches(SegFormer (Xie et al., 2021), Mask2Former (Cheng et al., 2022)) with semantic segmentation and panoptic segmentation (Xu et al., 2023a).

**Fully supervised cross-domain semantic segmentation** We evaluate the performance of our method on a cross-domain setting and adopt a baseline (Wei et al., 2023) for nighttime semantic segmentation on NightCity-fine (Tan et al., 2021).

Table 8: Closed set fully supervised semantic segmentation

| Methods | mIoU/PQ |
|---|---|
| SegFormer (B1) | 42.2 |
| +G4Seg | 42.9(+0.7) |
| Mask2Former(R50) | 47.2 |
| +G4Seg | 47.8(+0.6) |
| ODISE(panoptic) | 22.4 |
| +G4Seg | 23.0(+0.6) |

Table 9: Cross domain fully supervised semantic segmentation

| Methods | mIoU |
|---|---|
| DP (Wei et al., 2023) | 64.0 |
| DP+G4Seg | 64.5(+0.5) |

## I   SENSITIVITY ASSESSMENT ON COARSE MASK QUALITY BEFORE REFINEMENT

Ideally, our method does not rely on the initial mask quality. To show how sensitive the proposed method relying on the initial segmentation quality, since our approach is sample-wise, we performed stratification based on different quality levels of coarse segmentation and then calculated the mean IoU improvement for samples with different levels for the VOC dataset: When the initial mask quality

| Initial Mask Quality(IoU range) | 0-40 | 40-80 | 80-100 |
|---|---|---|---|
| # samples(#/# total samples) | 56(3.4%) | 679(47.3%) | 237(49.3%) |
| Avg G4Seg Gain | +0.2 | +1.9 | +1.1 |
| Avg Controlnet (Zhang et al.) Gain | +0.75 | +4.2 | +4.1 |
| Avg CascadedPSP (Cheng et al., 2020) Gain | +0.2 | +1.5 | +1.0 |

is poor, the improvement of our method is also limited. The improvement from our method is most significant for initial IoU values between 40 and 80. This indicates that our approach is particularly effective when the initial segmentation is already of reasonable quality. When the initial segmentation is already nearly perfect(80-100), the improvement from our method becomes limited due to the bottleneck caused by errors inherent in the mask injection process.

## J   RESULTS WITH OTHER MASK INJECTION METHOD

The overall pipeline of G4Seg is firstly obtaining a mask $S$ conditioned generative models $p(x|S)$ then updating the mask using the generative result with coarse mask. In first step, for serving the in-exact nature, we only use the attention perturbation in diffusion backbone avoiding involving exact pixel-level annotation.

Pursuing a better result with permission to use a pixel-level annotation, we could involving a more stronger mask injection method, Controlnet (Zhang et al.). The ControlNet consists of approximately half of a diffusion backbone and functions as a feature extractor that can accept arbitrary signals (such as segmentation masks) as input. The extracted features are then integrated into the diffusion backbone to control the generative output, $\epsilon(x_t, t, S)$. For images-annotation pairs($x_0$ and S), then the controlnet is trained with:

$$\mathcal{L}_{cn} = E_{\epsilon \sim N(0,I)}||\epsilon - \epsilon(x_t, t, S)||_2^2$$

For our implementations, we use the pretrained segmentation conditioned model provided by Lvmin Zhang & Agrawala which is then fine-tuned on the corresponding training set with the nearest palette defined by Lvmin Zhang & Agrawala. With an improved pipeline, the performance would significantly improve, as shown in the following table:

| Method | CLIP-ES VOC | SCLIP CoCo | SCLIP Context |
|---|---|---|---|
| G4Seg+EMI(Attn Injection) | 72.0 | 30.9 | 31.3 |
| G4Seg+EMI(ControlNet Injection) | 74.1 | 33.1 | 33.8 |

## K    COMPARISON WITH RELATED WORKS

**On-top-of**. Our method, as a plug-and-play framework, can be simply and efficiently integrated into various existing segmentation modules to enhance the performance online with the current single sample.

**Generative content with generated-original bias**. Some work such as VPD (Zhao et al., 2023) and ODISE (Xu et al., 2023a) use pretrained diffusion model as feature extractor with a self-supervised denoising loss. While another line of research, such as OVDiff (Karazija et al., 2023) and Freeda Barsellotti et al. (2024c), merely utilize the content directly generated by diffusion models for target prototype retrieval. In our work, we explore the discrepancy between the generative content and the initial image to refine the discriminative result.

**Discriminative assistance**.  Some diffusion-based training-free segmentation works such as Freeda (Barsellotti et al., 2024c) and OVDiff (Karazija et al., 2023) employ pre-trained discriminative models such as DINO (Caron et al., 2021) as assistance, while the performance of the framework heavily depends on these discriminative models.

**Cross attention initialization**. Most works employ the attention between text and image as a segmentation prior to the diffusion model, such as DatsetDiffusion (Nguyen et al., 2024) and DiffSegmentor (Wang et al., 2023b). The most significant difference between our work and others is that the attention mechanism we used is EMI as injecting the coarse mask prior to the generation pipeline. The EMI part could be substituted without attention with another more advanced mask-injecting module.

## L    G4SEG IMPLEMENTED WITH DIFFERENT DIFFUSION VERSION

We have compared the results with SD1.5, SD2.1, SDXL and LCM. SD1.5, SD2.1, LCM, and SDXL share largely similar U-Net backbone architectures, incorporating cross-attention and self-attention layers. Consequently, the EMI step is executed in a nearly identical manner across these models. Then after the generation, the SCA step remains the same.

| Diffusion Version | mIoU |
|---|---|
| SD1.5 | 71.8 |
| SD2.1 | 72.0 |
| SDXL | 72.0 |
| LCM | 72.1 |

## M    COMPARISON WITH TRAINING-FREE DIFFUSION SEGMENTATION METHODS

As the table shows, the DiffSegmentor only relies on the attention mechanism in the diffusion backbone as the clue to the target mask, which does not fully excavate the generation prior to the diffusion model. The OVDiff and Freeda utilize many generated images with a specific class and obtaining the discriminative prototype of the class, where the prototype is retrieved based on the region of interest from cross/Self-attention aggregation. Due to the strong external discriminative assistance, it is challenging to determine whether the generative capacity of the diffusion model contributes to the performance. Our method aims to fully exploit the generative prior for a discriminative task, specifically inexact segmentation, by adopting a GPT-like approach to solve the discriminative task in a generative manner without any extra assistance.

| Training-free Methods | Generative Content | Gen->Seg | discriminative assitance |
|---|---|---|---|
| OVDiff[1] | Class conidtioned images | Cross/Self attention | discriminative feature prototype |
| Freeda[2] | Class conidtioned images | Cross/Self attention | discriminative feature prototype |
| DiffSegmentor[3] | None | Cross/Self attention | None |
| G4Seg | Mask conidtioned images | Semantic correspondence updating | None |

# N    More Results on Comparison with Other Mask Refinement Methods

We have also conducted a comparison between other mask refinement methods on VOC and Context datasets with SCLIP and MaskCLIP.

| Methods | SCLIP VOC | SCLIP Context | MaskCLIP VOC | MaskCLIP Context |
|---|---|---|---|---|
| Baseline | 59.1 | 30.4 | 38.8 | 23.6 |
| +G4Seg | 59.8(+0.7) | 31.3(+0.9) | 39.4(+0.6) | 24.1(+0.5) |
| +SegRefiner | 59.3(+0.2) | 30.7(+0.3) | 39.1(+0.3) | 23.9(+0.3) |
| +CascadePSP | 59.5(+0.4) | 30.9(+0.5) | 39.2(+0.4) | 23.8(+0.2) |
| +Densecrf | 60.9(+1.8) | 31.2(+0.8) | 39.9(+1.1) | 24.2(+0.6) |
| +G4Seg + CascadePSP | 60.1(+1.0) | 31.6(+1.2) | 39.5(+0.7) | 24.3(+0.7) |
| +G4Seg+Densecrf | **62.1(+3.0)** | **32.0(+1.6)** | **40.1(+1.3)** | **24.6(+1.0)** |

# O    More Visualized Results

## O.1    Dataset Details

**Datasets.** In Section 4, we evaluate our G4Seg on 3 prevalent benchmarks, which are PASCAL VOC12 2012 Everingham et al. (2015), COCO Lin et al. (2014), PASCAL Context Mottaghi et al. (2014). Here is the detailed introduction of these five datasets as follows:

- **PASCAL VOC2012 Everingham et al. (2015):** The PASCAL VOC12 dataset consists of a diverse collection of images spanning 21 different object categories (including one background class), such as a person, car, dog, and chair. The dataset provides annotations for both training and validation sets, with around 1,464 images in the training set and 1,449 images in the validation set. We use the validation set for the downstream evaluation.

- **COCO Lin et al. (2014):** The COCO Object dataset covers a wide range of 80 object categories, such as cars, bicycles, people, animals, and household items. For semantic segmentation, it has 118,287 training images and 5,000 images for validation.

- **Context Mottaghi et al. (2014):** The dataset contains a diverse set of images taken from various scenes, including indoor and outdoor environments. It covers 59 common object classes, such as a person, car, bicycle, and tree, as well as 60 additional stuff classes, including sky, road, grass, and water. It has 118,287 training images and 5,000 images for validation. Here we merely consider the object dataset part and use the validation set.

## O.2 VOC RESULTS

Figure 10 presents more results of our G4Seg in VOC12. It is found that our G4Seg shows powerful grouping capability when segmenting the object-centric images. Besides, the generated discrepancy could help segment objects in a compact and dense manner, which means there is less redundancy and noise in objects.

## O.3 COCO RESULTS

Figure 11 presents some visualized results of COCO Object. Clearly, it has been observed that, compared to GroupViT, our G4Seg is able to perform fine-grained segmentation in the multi-object case. However, G4Seg is unable to provide full areas of object segmentation, revealing the bottleneck of our method.

## O.4 CONTEXT RESULTS

Figure 12 shows several visualized results of Context. A similar improvement could be observed. Besides, G4Seg could enhance the discriminative regions to a large extent in some cases, indicating its effectiveness in multi-object learning.

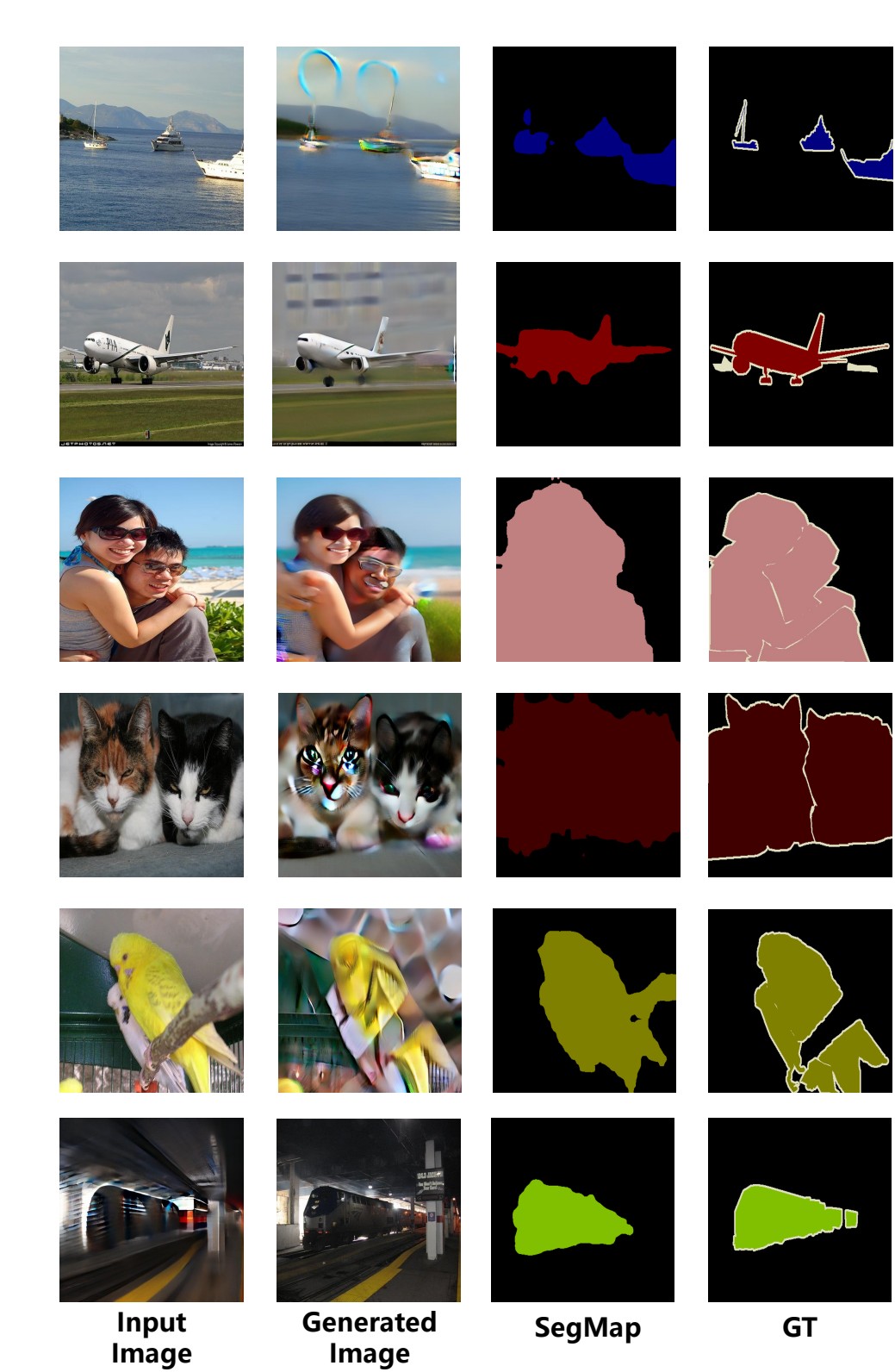

**Input Image** **Generated Image** **SegMap** **GT**

Figure 10: Qualitative results on PASCAL VOC12.

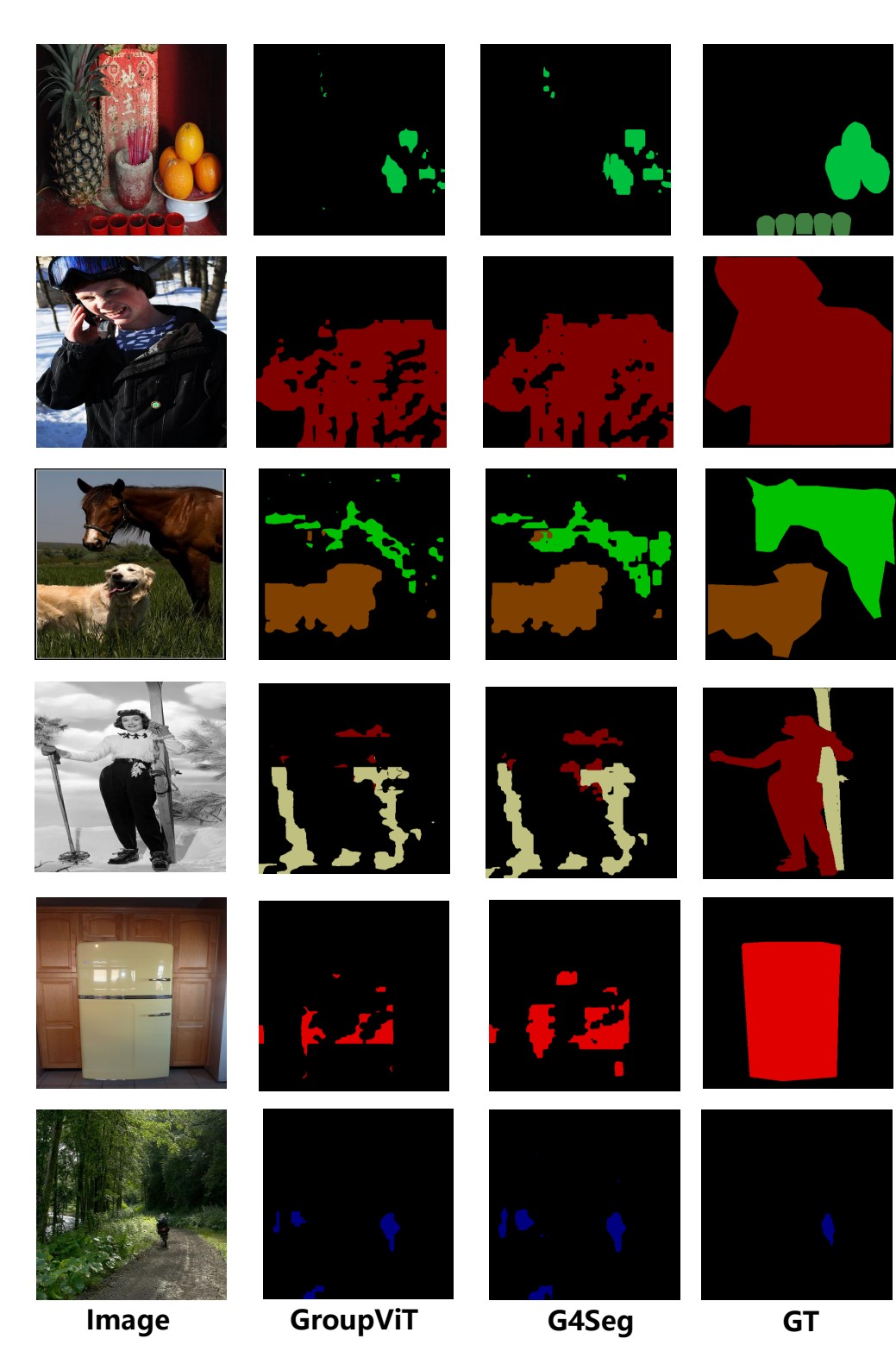

Figure 11: Qualitative results on COCO Object.

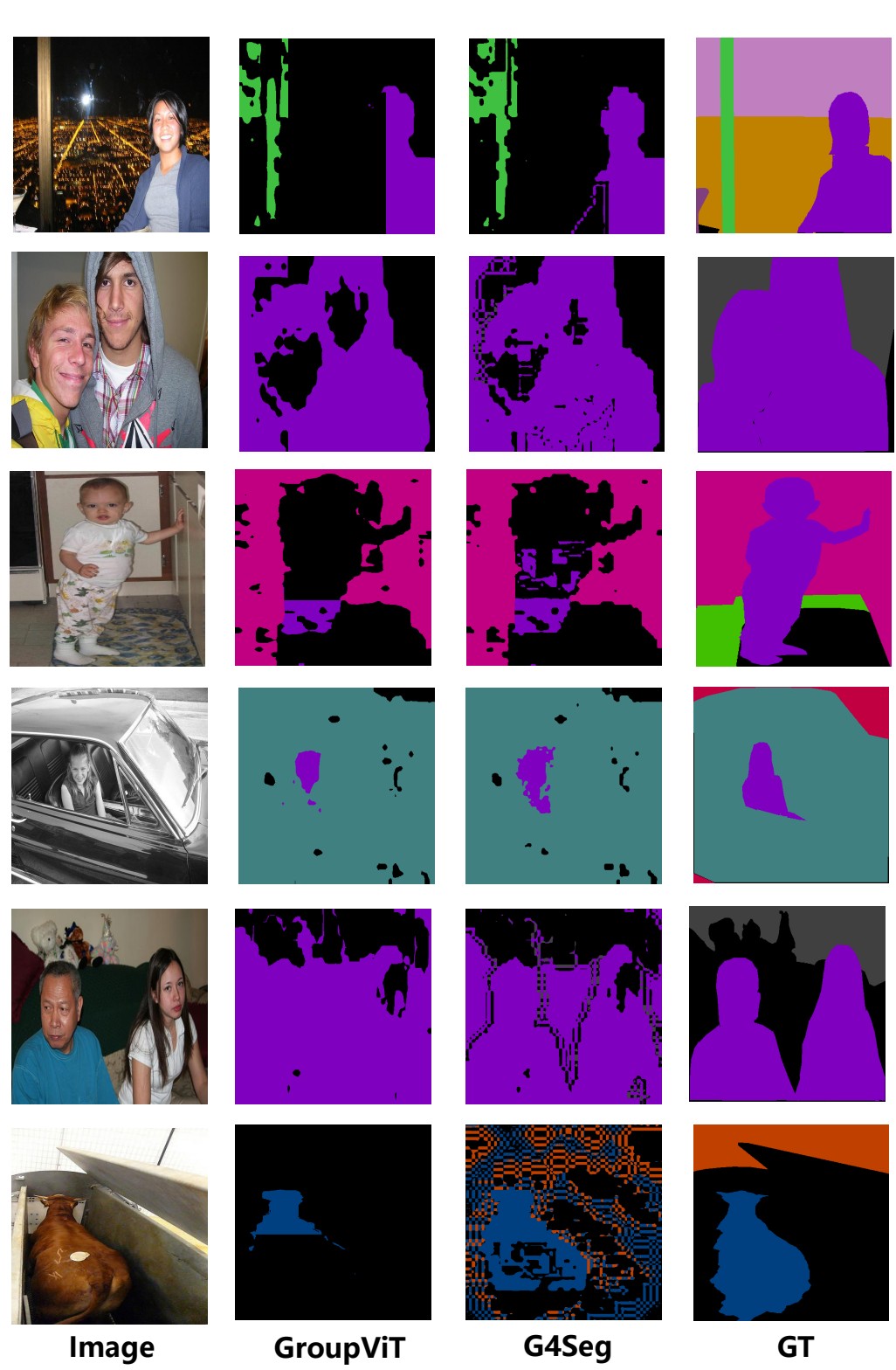

Figure 12: Qualitative results on Context.

