# OpenReview forum: "G4Seg: Generation for Online Segmentation Refinement with Diffusion Models"
_ICLR.cc/2025/Conference — ICLR 2025 Conference Withdrawn Submission_

### Official Review · Reviewer_XTz2 · 2024-10-30

**Soundness:** 3
**Presentation:** 2
**Contribution:** 2
**Rating:** 5
**Confidence:** 4

**Summary:**

In this paper, the authors  exploits the discrepancies between original images and coarse mask-conditional generated images to improve mask quality. The proposed method has performance gain on both open-vocabulary and weakly-supervised task.

**Strengths:**

(1) The motivation of reducing discrepancies between original images and coarse mask-conditional generated images is easy for understanding.

(2) The proposed method can improve the segmentation quality of existing open-vocabulary and weakly-supervised methods.

**Weaknesses:**

(1) The contribution of this paper is limited, and the idea of reducing discrepancies is relatively incremental.

(2) The proposed method focuses on discrepancies. For irregular objects, it is difficult to give an accurate generation. Does the proposed method have negative results on these situations.

(3) The proposed method has limited improvement by performing a forward diffusion denoising process.

(4) The proposed method is a refinement processing, which lacks of comparison with existing mask refinement method, such as CRF.

(5) It lacks the analysis about the hyper-parameters like injection weight and mixing coefficient. For implementation details, the settings on different tasks are different.

(6) The proposed method seems increasing the computational cost during inference according to Table 5.

**Questions:**

(1) The contribution and novelty could be improved

(2) More discussions about the discrepancies, and more experimental comparison

---

> ### Author Response · Authors · 2024-11-21
> **Response to Reviewer XTz2 1/3**
>
> ### W1 &  Q1
>
> >The contribution of this paper is limited, and the idea of reducing discrepancies is relatively incremental.
> >The contribution and novelty could be improved
> ###
>
> **Limited Contribution**
> We respectfully disagree with this. We'd like to argue that the distinctive aspect of our G4Seg, is the first work to solve the discriminative segmentation task with **dynamic generative content mining, which is based on an interesting finding regarding the effectiveness of the generated-original visual bias on the segmentation mask**. In this way, we make contributions on performing a **plug-and-play training-free** refinement framework with the pre-trained diffusion models leveraging the generation pior. We have made a comparison with other segmentation approaches utilizing the diffusion model in Table 1 in the revised manuscript. And a more detailed table compares training-free diffusion-based semantic segmentation methods in terms of the generative content, the strategy mimic generation and segmentation gap, and external discriminative assistance.
>
> | Training-free Methods | Generative Content       | Gen->Seg                         | Discriminative assitance         |
> | --------------------- | ------------------------ | -------------------------------- | -------------------------------- |
> | OVDiff                | Class conidtioned images | Cross/Self attention             | Discriminative feature prototype |
> | Freeda                | Class conidtioned images | Cross/Self attention             | Discriminative feature prototype |
> | DiffSegmentor         | None                     | Cross/Self attention             | None                             |
> | G4Seg                 | Mask conidtioned images  | Semantic correspondence updating | None                             |
>
> In addition to re-claiming our contribution to the methodology, G4Seg also consistently improves the performance **both on in-exact segmentation and fully/semi-supervised segmentation** via leveraging the generation of prior knowledge.
>
> For **fully/semi-supervised open-vocabulary segmentation**:
> | Methods | VOC             | Context        |
> | ------- | --------------- | -------------- |
> | OVAM    | 61.2            | 28.3           |
> | +G4Seg  | **62.1(+0.9)**  | **28.9(+0.6)** |
> | DeOP    | 91.7            | 48.8           |
> | +G4Seg  | **92.1 (+0.4)** | **49.3(+0.5)** |
>
> For **fully supervised closed set sematic segmentation/panoptic segmentation**:
> | Methods          | mIoU/PQ        |
> | ---------------- | -------------- |
> | SegFormer (B1)   | 42.2           |
> | +G4Seg           | **42.9(+0.7)** |
> | Mask2Former(R50) | 47.2           |
> | +G4Seg           | **47.8(+0.6)** |
> | ODISE(panoptic)  | 22.4           |
> | +G4Seg           | **23.0(+0.6)** |
>
> We have added these to the Appendix of the revised paper.
>
> **Incremental contribution for reducing discrepancies**
>
> Here, we want to emphasize that there is an **inherent gap between the generative prior and the segmentation results**. For a diffusion model, its training objective is to progressively denoise an image step-by-step from noise. The aforementioned concurrent works are also trying to address this gap, but they overlook the capabilities of generating specific images of the diffusion model. Therefore, we **aim to leverage a generative nature of the diffusion model with dynamic generative content mining to tackle a downstream pixel-level discrimination task**, without relying heavily upon external discriminative assistance.

---

> ### Author Response · Authors · 2024-11-21
> **Response to Reviewer XTz2 2/3**
>
> ### W2 & W3 & Q2
> >The proposed method focuses on discrepancies. For irregular objects, it is difficult to give an accurate generation. Does the proposed method have negative results in these situations?
> >The proposed method has limited improvement by performing a forward diffusion denoising process.
> >More discussions about the discrepancies, and more experimental comparison.
>
> **limited improvement and irregular condition**
> We agree with the generation limitation. In some extreme cases, when the object is in irregular shape and initial mask is of poor quality, then the improvements become limited.
> We stratified the samples based on different quality levels of coarse segmentation and calculated the **mean IoU improvement for each group according to their initial mask quality**, as measured by the per-sample IoU metric on the VOC dataset:
>
>
>
> |Initial Mask Quality(IoU range)       | 0-40     | 40-80      | 80-100     |
> | ------------------ | -------- | ---------- | ---------- |
> |# samples(#/# total samples)         | 56(3.4%) | 679(47.3%) | 237(49.3%) |
> | Avg G4Seg          | +0.2     | +1.9       | +1.1       |
> | Avg ControlNet[7]  | +0.75    | +4.2       | +4.1       |
> | Avg CascadedPSP[8] | +0.2     | +1.5       | +1.0       |
>
> *ControlNet means Explicit Mask Injection using a pre-trained Controlnet*
> As seen from the Table, we make the following comments:
> - **Irregular shape and bad initial mask**. For initial coarse mask for relatively low performance,  all three methods(including one fully supervised method) have limited improvements compared to higher initial segmentation.
> - **Limited Improvement** Since we have introduced a mask conditioning method which is based on cross-attention and self-attention, is the performance bottleneck. These attention-based generation methods do not perform well in mask-conditioned generation. If we adopt a stronger mask conditioning method, such as Contronet[7], the performance would significantly improve, as shown in the Table.
>
> *We do not adopt such a stronger mask injection strategy because it would violate the "inexact segmentation" setting in our approach*.
>
> **Discussion of the discrepancies**
> In addition to the theoretical explanations in the main text, we have included more **intuitive explanations, as shown in the Appendix F** . Furthermore, we found that the primary bottleneck for G4Seg in leveraging discrepancies lies in the method of mask injection. As demonstrated in the table above, if we adopt **a more precise mask injection approach, our method's utilization of discrepancies becomes more effective and reasonable**.
>
> We have added these discussions to the revised paper with Appendix I.
>
>
> ### W4
> >The proposed method is a refinement processing, which lacks of comparison with existing mask refinement method, such as CRF.
>
> Sorry for the missing comparison.  We have conduct a comparison between other mask refinement method on VOC and Context datasets with SCLIP and MaskCLIP.
> | Methods             | SCLIP VOC      | SCLIP Context  | MaskCLIP VOC   | MaskCLIP Context |
> | ------------------- | -------------- | -------------- | -------------- | ---------------- |
> | Baseline            | 59.1           | 30.4           | 38.8           | 23.6             |
> | +G4Seg              | 59.8(+0.7)     | 31.3(+0.9)     | 39.4(+0.6)     | 24.1(+0.5)       |
> | +SegRefiner         | 59.3(+0.2)     | 30.7(+0.3)     | 39.1(+0.3)     | 23.9(+0.3)       |
> | +CascadePSP         | 59.5(+0.4)     | 30.9(+0.5)     | 39.2(+0.4)     | 23.8(+0.2)       |
> | +Densecrf           | 60.9(+1.8)     | 31.2(+0.8)     | 39.9(+1.1)     | 24.2(+0.6)       |
> | +G4Seg + CascadePSP | 60.1(+1.0)     | 31.6(+1.2)     | 39.5(+0.7)     | 24.3(+0.7)       |
> | +**G4Seg+Densecrf** | **62.1(+3.0)** | **32.0(+1.6)** | **40.1(+1.3)** | **24.6(+1.0)**   |
>
> According to the Table, we make the following comments:
> - Although CascadePSP and SegRefiner use many pixel-level labels for training, the performance improvement in exact semantic segmentation is still quite limited.
> - DenseCRF, as a method that refines coarse predictions by leveraging the information of image formation, improves the initial segmentation with a significant margin when the number of classes is limited. However, as the number of classes increases, the improvements achieved by DenseCRF become less significant. The improvements of our method are **roughly comparable to those of DenseCRF on Context dataset**.
> - Since the source of segmentation knowledge in our method differs from that of other approaches (CascadePSP relies on annotations, and DenseCRF leverages image-based priors), G4Seg can be **further combined with these methods to achieve additional improvements**.

---

> ### Author Response · Authors · 2024-11-21
> **Response to Reviewer XTz2 3/3**
>
> ### W5
> >It lacks the analysis about the hyper-parameters like injection weight and mixing coefficient. For implementation details, the settings on different tasks are different.
>
> **Missing ablation study**
> Sorry for the missing experiments. We add these experiemnts following with different injection values and different  mixing coefficient.
>
> **Performance on VOC with different injection values**
> | Hyper params / mIoU on VOC| 0.5  | 1.0  | 2.0  | 5.0  | 7.0  | 10.0 |
> | - | - | - | ---- | - | - | - |
> | $\alpha$ injection val (5) | 71.9 | 72.0 | 72.0 | 71.9 | 71.5 | 71.3 |
>
> So we finally choose injection value 1.0.
> Then the mixing coefficient ablation for weakly supervised semantic segmentation is listed as follows:
> **Performance on WSSS VOC with different mixing coefficients**
> | Hyper params / mIoU on VOC| 0.1  | 0.3  | 0.5  | 0.7  | 0.9  |
> | - | - | - | ---- | - | - |
> | mixing weight $\beta$ | 70.1 | 70.6 | 71.2 | 71.9 | 72.0 |
>
> The ablation for text-supervised semantic segmentation on VOC is listed in the following:
> | Hyper params / mIoU on VOC | 0.35 | 0.5  | 0.65 | 0.8  | 0.85 | 0.9  | 0.95 |
> | -------------------------- | ---- | ---- | ---- | ---- | ---- | ---- | ---- |
> | mixing weight $\beta$      | 58.1 | 59.2 | 59.3 | 59.5 | 59.8 | 59.6 | 59.4 |
>
>
> **Different settings**
> We'd like to emphasize that our experiment settings **remain the same for both weakly supervised semantic segmentation and text-supervised semantic segmentation**. Our approach utilizes existing base methods to generate a coarse mask, which is then refined using a pre-trained diffusion model. For different experiments, we use different hyperparameters. The widely used DenseCRF follows a similar procedure[9].
>
> The $\beta$ coefficient represents **the reliance of initial segmentation confidence**. A lower mixing weight is chosen for text-supervised semantic segmentation since the base segmentation performance of TSSS is relatively poor, requiring us to rely more heavily on the corresponding component for further refinement.
>
>
>
> ### W6
> >The proposed method seems increasing the computational cost during inference according to Table 5.
>
> We appreciate the authors' constructive criticism. As a method that leverages the **generative capabilities of diffusion models** for post-processing **without any annotation**, it is natural for our approach to incur additional computational overhead. However, after comparing the computational latency of several mask refinement methods, we found that our approach **operates within a reasonable range of computational cost**. Finally, the computational cost of diffusion models could be improved with pruning, quantization, or using lighter versions of Stable Diffusion models.
>
> | method           | Inference Speed |
> | ---------------- | --------------- |
> | +G4Seg           | +1.3s           |
> | +Dense CRF       | +0.9s           |
> | +Slide Inference | +1.8s           |
> | +SegRefiner      | +2.0s           |
>
> *SegRefiner is a fully supervised mask refinement method with a diffusion process*
>
>
>
>
>
>
> >[1] Unleashing text-to-image diffusion models for visual perception. ICCV2023.
>
> >[2] Open-vocabulary panoptic segmentation with text-to-image diffusion models. CVPR2023.
>
> >[3] Diffusion models for zero-shot open-vocabulary segmentation. arXiv:2306.09316.
>
> >[4] Diffusion model is secretly a training-free open vocabulary semantic segmenter. arXiv:2309.02773.
>
> >[5] Training-Free Open-Vocabulary Segmentation with Offline Diffusion-Augmented Prototype Generation. CVPR2024.
>
> >[6] Dataset diffusion: Diffusion-based synthetic data generation for pixel-level semantic segmentation. Neurips2023.
>
> >[7] https://huggingface.co/lllyasviel/sd-controlnet-seg
>
> >[8] CascadePSP: Toward Class-Agnostic and Very High-Resolution Segmentation via Global and Local Refinement. CVPR2020.
>
> >[9] https://github.com/lucasb-eyer/pydensecrf/issues/76

---

> > ### Author Response · Authors · 2024-11-24
> >
> > Dear Reviewer XTz2:
> >
> > Thank you once again for taking the time to review our work and providing insightful feedback. We sincerely hope our responses have clarified the points raised. Should you have any additional questions or concerns, please feel free to reach out for further discussion.
> >
> > Best regards,
> >
> > Authors of Submission 6302

---

> > > ### Comment · Reviewer_XTz2 · 2024-11-27
> > >
> > > I have checked the author's response and other reviews. Some concerns have been addressed. I still feel that the current method offers limited contribution but introduces significant complexity.

---

### Official Review · Reviewer_a25d · 2024-10-31

**Soundness:** 3
**Presentation:** 3
**Contribution:** 2
**Rating:** 5
**Confidence:** 3

**Summary:**

This paper proposes an effective method G4Seg based on Stable Diffusion to tackle the Inexact Segmentation task. This method utilizes the difference between the generated and original image to help refine the mask. By establishing a semantic correspondence alignment, the foreground probability of the confusion area can be effectively updated. Experiments on various benchmarks validate the effectiveness and superiority of this training-free approach.

**Strengths:**

(1) The idea that using the difference between the generated and original image to help refine the mask is interesting.

(2) This paper is generally well-written and easy to follow.

(3) The introduction of G4Seg is comprehensive and detailed.

(4) G4Seg is training-free and easy-to-use.

(5) The framework is tested across multiple benchmarks, including TSSS and WSSS, showcasing its versatility and effectiveness under different settings.

**Weaknesses:**

(1) G4Seg is based on Stable Diffusion and suffers from heavy time costs, resulting in poor practicality especially when inference at scale.  It would be better to report the specific inference cost for each dataset, not be confined to VOC but the complex COCO.

(2) As shown in the figures, the refinement of confusion areas is limited and imperfect. There are still a certain amount of pixels that G4Seg can not correct. What is the inherent reason for this phenomenon? The struggles to select confusion areas or failures in semantic correspondence alignment? Meanwhile, G4Seg should be compared to other mask refinement techniques such as training-free dense CRF and training-based CascadePSP [1], SegRefiner[2]. It seems that dense CRF is superior and more efficient than G4Seg. For training-based methods, I understand that these works may use additional pixel-level annotations for training, but the comparison can help readers understand the gap between G4Seg and SOTA mask refinement techniques.

(3) The feature maps in Stable Diffusion are heavily downsampled (e.g., 64x64). Does G4Seg work for small objects?

(4) VOC is not an ideal dataset to validate the boundary accuracy because areas near the boundary are labeled as “void” in GT masks. Datasets with more boundary-accurate annotations, like re-labeled VOC and BIG used in CascadePSP [1], are more appropriate.

(5) Detailed parameter sensitivity analyses for the injection weight $\alpha$ and mixing coefficient $\beta$ are lost, which is important to assess the robustness of G4Seg.

---

[1] CascadePSP: Toward Class-Agnostic and Very High-Resolution Segmentation via Global and Local Refinement.

[2] SegRefiner: Towards Model-Agnostic Segmentation Refinement with Discrete Diffusion Process.

**Questions:**

In Table 6, how to inject boxes, points and scribbles into Stable Diffusion?

---

> ### Author Response · Authors · 2024-11-21
> **Response to Reviewer a25d 1/2**
>
> ### W1
> >G4Seg is based on Stable Diffusion and suffers from heavy time costs, resulting in poor practicality especially when inference at scale. It would be better to report the specific inference cost for each dataset, not be confined to VOC but the complex COCO.
>
>
> **inference cost on CoCo**
>
> | Seg   | inference speed VOC | inference speed on COCO |
> | ----- | ------------------- | ----------------------- |
> | G4Seg | +1.1 s/sample       | 1.6s/sample             |
>
> Since our method updates one segment with one pass, and the COCO dataset contains an average of multiple categories per image, the overall processing time per image is relatively longer.
>
> **Inference cost comparison with other refinement methods**
> After comparing the average inference speed on three datasets of several post-hoc mask refinement methods, we found that our approach operates within a reasonable range of computational costs.
> | method           | Inference Speed |
> | ---------------- | --------------- |
> | +G4Seg           | +1.3s           |
> | +Densecrf        | +0.9s           |
> | +Slide Inference | +1.8s           |
> | +SegRefiner[2]   | +1.7s           |
>
> ### W3
> > The feature maps in Stable Diffusion are heavily downsampled (e.g., 64x64). Does G4Seg work for small objects?
>
>
> Since the feature maps are downsampled, as for extremely small objects, we can use a diffusion model with a larger input resolution such as SDXL(128x128 feature map size). Furthermore, resizing the image to a larger resolution and then applying sliding inference also help to detect small targets. We conduct these two strategies on the VOC dataset and observe the performance on **small object, which is defined with total occupied object pixels is smaller than 10% of the total pixels in the image**.
> | Methods       | mIoU for small object(<10% pixels) |
> | ------------- | ---------------------------------- |
> | SCLIP         | 30.2                               |
> | +G4Seg        | 30.7                               |
> | +G4Seg(SDXL)  | 30.9                               |
> | +G4Seg(Slide) | **31.0**                               |
>
>
>
>
> ### W5
> > Detailed parameter sensitivity analyses for the injection weight and mixing coefficient are lost, which is important to assess the robustness of G4Seg.
>
> Sorry for this. Based on the following table, as the increase of injection value (in equations (5) and (6)), the performance meets a **initial increase followed by a subsequent decrease**. Therefore, we choose 1.0 as our hyperparameter to achieve the best numerical value. We will add this.
>
> **Performance on VOC with different injection values**
> | Hyper params / mIoU on VOC| 0.5  | 1.0  | 2.0  | 5.0  | 7.0  | 10.0 |
> | - | - | - | ---- | - | - | - |
> |  injection val $\alpha$ (5) | 71.9 | 72.0 | 72.0 | 71.9 | 71.5 | 71.3 |
>
> Then the mixing coefficient ablation is following:
> **Performance on VOC with different mixing coefficient**
> | Hyper params / mIoU on VOC| 0.1  | 0.3  | 0.5  | 0.7  | 0.9  |
> | - | - | - | ---- | - | - |
> | mixing weight $\beta$ | 71.5 | 71.6 | 71.6 | 71.9 | 72.0 |
>
>
> ### Q1
> > In Table 6, how to inject boxes, points and scribbles into Stable Diffusion?
>
> Sorry for this. Actually, in the test phase, a coarse mask is obatined firstly based on these visual prompt via the specific segmentation method. Then, G4Seg is applied on top of it. We have modified paper to clarify it.

---

> > ### Author Response · Authors · 2024-11-21
> > **Response to Reviewer a25d 2/2**
> >
> > ### W2 & W4
> > >As shown in the figures, the refinement of confusion areas is limited and imperfect. There are still a certain amount of pixels that G4Seg can not correct. What is the inherent reason for this phenomenon? The struggles to select confusion areas or failures in semantic correspondence alignment? Meanwhile, G4Seg should be compared to other mask refinement techniques such as training-free dense CRF and training-based CascadePSP [1], SegRefiner[2]. It seems that dense CRF is superior and more efficient than G4Seg. For training-based methods, I understand that these works may use additional pixel-level annotations for training, but the comparison can help readers understand the gap between G4Seg and SOTA mask refinement techniques.
> > > VOC is not an ideal dataset to validate the boundary accuracy because areas near the boundary are labeled as “void” in GT masks. Datasets with more boundary-accurate annotations, like re-labeled VOC and BIG used in CascadePSP [1], are more appropriate.
> >
> > **Pixels, not Correct**
> > Our method appears to correct the edges. However, due to the inherent errors in the mask injection process(based on perturbing the attention in the diffusion backbone), our edge corrections are not as precise as methods specifically designed with edge priors with annotation.
> > If we adopt stronger mask injection methods, such as Controlnet[3], the performance is expected to improve further, as shown in the following Table:
> >
> > | Method                          | CLIP-ES VOC | SCLIP CoCo | SCLIP Pascal Context |
> > | --- | ---- | --- | --- |
> > | G4Seg+EMI(Attn Injection)    | 72.0   | 30.9  | 31.3     |
> > | G4Seg+EMI(Controlnet Injection) | 74.1  | 33.1  | 33.8   |
> >
> > **Other Mask Refinement Methods**
> > As mentioned, CascadePSP and SegRefiner focus on improving segmentation and require pixel-wise annotations for training. As for semantic segmentation, **these methods may focus more on improving segmentation around the boundary**. Dense CRF is a widely used traditional method that leverages priors constructed from the image itself to refine the coarse mask. In contrast, our approach emphasizes **adapting the generative capabilities of diffusion models to refine in-exact segmentation problems**. We have conducted a comparison between these methods on VOC and Context datasets on top of SCLIP and MaskCLIP.
> > | Methods             | SCLIP VOC      | SCLIP Context  | MaskCLIP VOC   | MaskCLIP Context |
> > | ------------------- | -------------- | -------------- | -------------- | ---------------- |
> > | Baseline    | 59.1       | 30.4      | 38.8     | 23.6       |
> > | +G4Seg       | 59.8(+0.7)     | 31.3(+0.9)     | 39.4(+0.6)     | 24.1(+0.5)       |
> > | +SegRefiner    | 59.3(+0.2)     | 30.7(+0.3)     | 39.1(+0.3)     | 23.9(+0.3)       |
> > | +CascadePSP   | 59.5(+0.4)     | 30.9(+0.5)     | 39.2(+0.4)     | 23.8(+0.2)       |
> > | +Densecrf           | 60.9(+1.8)     | 31.2(+0.8)     | 39.9(+1.1)     | 24.2(+0.6)       |
> > | +G4Seg + CascadePSP | 60.1(+1.0)     | 31.6(+1.2)     | 39.5(+0.7)     | 24.3(+0.7)       |
> > | +**G4Seg+Densecrf** | **62.1(+3.0)** | **32.0(+1.6)** | **40.1(+1.3)** | **24.6(+1.0)**   |
> >
> > According to the Table, we make the following comments:
> > - Although CascadePSP and SegRefiner use many pixel-level labels for training, the performance improvement in in-exact semantic segmentation is still quite limited.
> > - DenseCRF, as a method that refines coarse predictions by leveraging the information of image formation, improve the initial segmentation with a significant margin when the number of classes is limited. However, as the number of classes increases, the improvements achieved by DenseCRF become less significant. The improvements of our method are **roughly comparable to those of DenseCRF on Context dataset**.
> > - Since the source of segmentation knowledge in our method differs from that of other approaches (CascadePSP relies on annotations, and DenseCRF leverages image-based priors), **G4Seg can be further combined with these methods to achieve additional improvements**.
> >
> > **Experiments on Boundary Enhancement Dataset**
> > We have conducted our G4Seg on relabeled Pascal VOC dataset:
> > | Methods                                            | mIoU        |
> > | -------------------------------------------------- | ----------- |
> > | PSPNet                                             | 90.4        |
> > | +CascadePSP(Training Under Pixel Level Annotation) | 93.9 (+3.5) |
> > | +G4Seg (training-free)                             | 91.1 (+0.7) |
> >
> > Note that the relabeled Pascal VOC dataset only contains 500 images with almost one class per image. For the initial mask with high precision, there is a clear gap between our method and the fully supervised method.
> >
> > >[1] CascadePSP: Toward Class-Agnostic and Very High-Resolution Segmentation via Global and Local Refinement. CVPR2020.
> >
> > >[2] SegRefiner: Towards Model-Agnostic Segmentation Refinement with Discrete Diffusion Process. NeurIPs2023.
> >
> > >[3] https://huggingface.co/lllyasviel/sd-controlnet-seg

---

> > > ### Comment · Reviewer_a25d · 2024-11-25
> > >
> > > Thanks for the authors’ responses, which have resolved some of my concerns. However, I still have a few remaining questions:
> > >
> > > 1. My primary concern is the marginal performance improvement of this method. It seems that the performance gain on different sources of coarse masks is relatively small (typically less than 1%), which seems to contradict the claim that the generative capabilities of diffusion models can be used to refine inexact segmentation problems. The limited improvement suggests that the intrinsic generative priors in diffusion models may not be well-suited for this task.
> > >
> > > 2. As mentioned, the inaccurate results are attributed to the inherent errors in the mask injection process, and it was suggested that stronger mask injection methods like ControlNet could enhance performance. Could the authors elaborate on how ControlNet can be applied in this setting? I believe this solution could be highly valuable for achieving more effective refinement.
> > >
> > > 3. It would also be helpful to explain the details of how boxes, points, and scribbles are injected into Stable Diffusion, such as the specific segmentation method used, to ensure reproducibility.
> > >
> > > 4. From my own experience working on the mask refinement task, I noticed that some of the baseline results provided appear to be unexpectedly low and inconsistent with my reproductions. For instance, in my experiments, CascadePSP improved the performance of CLIP-ES from 70.8 to around 77, whereas in the results presented, it only achieved a slight improvement on different settings, which is strange.
> > >
> > > I would greatly appreciate it if the authors could address these concerns.

---

> > > > ### Author Response · Authors · 2024-11-25
> > > > **Response to Reviewer a25d Round2 1/2**
> > > >
> > > > ### Q1
> > > > > My primary concern is the marginal performance improvement of this method. It seems that the performance gain on different sources of coarse masks is relatively small (typically less than 1%), which seems to contradict the claim that the generative capabilities of diffusion models can be used to refine inexact segmentation problems. The limited improvement suggests that the intrinsic generative priors in diffusion models may not be well-suited for this task.
> > > >
> > > > We make following clarification:
> > > > 1. We argue that our method **achieves improvements across a variety of approaches, including WSSS, TSSS, as well as fully and semi-supervised methods**. For the CascadePSP trained with pixel-level annotations reviewer mentioned on a **more complex multi-class segmentation tasks**, the method also achieves less improvements(less than 1%) as shown in Table 4(also the following Table) from the CascadePSP paper.
> > > >
> > > > 2. Furthermore, under a more **strong mask injection methods**, our improvements could be more stronger. We have conducted our method based on ADE20k, and the final results are shown in the following Table. We have achieved **1.0% mIoU gain compared with CascadePSP(0.8%)**.
> > > >
> > > >
> > > > 3. We argue that **the intrinsic generative priors in diffusion models could be properly utilized for segmentation tasks**. It is not only the finding of our method; **a number of approaches are exploring how diffusion models can enhance segmentation tasks**. The generative prior has been **widely validated as effective for semantic segmentation**, as demonstrated by methods such as ODISE, VPD, DiffSegmentor, and DatasetDiffusion. The key difference between our approach and these methods lies in the **paradigm used to exploit the diffusion model's prior**.
> > > >
> > > > | Methods (on ADE20k)                     | RefineNet | EncNet   | PSPNet   | avg     |
> > > > | --------------------------------------- | --------- | -------- | -------- | ------- |
> > > > | +CascadePSP(from paper)                 | +0.7      | +1.0     | +0.7     | 0.8     |
> > > > | +G4Seg base(w/o pixel-level annotation) | +0.6      | +0.5     | +0.5     | 0.5     |
> > > > | +G4Seg Controlnet                       | +**1.1**  | +**1.2** | +**0.9** | **1.0** |
> > > >
> > > >
> > > > ### Q2
> > > > > As mentioned, the inaccurate results are attributed to the inherent errors in the mask injection process, and it was suggested that stronger mask injection methods like ControlNet could enhance performance. Could the authors elaborate on how ControlNet can be applied in this setting? I believe this solution could be highly valuable for achieving more effective refinement.
> > > >
> > > > Thank you for the advice. The overall pipeline of G4Seg is firstly obtaining a mask $S$ conditioned generative models $p(x|S)$ then updating the mask using the generative result with coarse mask. In first step, for serving the in-exact nature, we only use the attention perturbation in diffusion backbone avoiding involving exact pixel-level annotation.
> > > >
> > > > Persuing a better result with permission of using a pixel-level annotation, we could invoving a more **stronger mask injection method, Controlnet[1]**. The ControlNet consists of approximately half of a diffusion backbone and **functions as a feature extractor that can accept arbitrary signals (such as segmentation masks) as input. The extracted features are then integrated into the diffusion backbone to control the generative output, $\epsilon(x_t,t,S)$**. For images-anntation pairs($x_0$ and S), then the controlnet is trained with:
> > > >
> > > > $$
> > > > \mathbb{L} = E_{\epsilon \sim N(0,I)}||\epsilon-\epsilon(x_t,t,S)||_2^2
> > > > $$
> > > >
> > > > For our implementations, we use the pretrained segmentation conditioned model provided by [2] which is then finetuned on the corresponding training set with nearest palette defined by [2].
> > > >
> > > >
> > > >
> > > > [1] Adding Conditional Control to Text-to-Image Diffusion Models. ICCV2023
> > > >
> > > > [2] https://huggingface.co/lllyasviel/sd-controlnet-seg
> > > >
> > > > We have added these discussions in Appendix J.

---

> > > > ### Author Response · Authors · 2024-11-25
> > > > **Response to Reviewer a25d Round2 2/2**
> > > >
> > > > ### Q3
> > > > > It would also be helpful to explain the details of how boxes, points, and scribbles are injected into Stable Diffusion, such as the specific segmentation method used, to ensure reproducibility.
> > > >
> > > > Sorry for the misleading again. We make further clarifications and show following pipeline:
> > > > **Box/scribbles/Points**--baseline(SPML)-->**Coarse Mask--->Our Pipeline**
> > > >
> > > > Boxes, points, and scribbles are also forms of inexact weak labels compared to text or image labels. These labels are **not directly injected into the diffusion models**. Instead, a coarse mask is first obtained from these labels using corresponding weakly supervised semantic segmentation methods, and **this coarse mask is then injected into the Stable Diffusion**.
> > > >
> > > > We have added these into the revision paper.
> > > >
> > > > ### Q4
> > > > > From my own experience working on the mask refinement task, I noticed that some of the baseline results provided appear to be unexpectedly low and inconsistent with my reproductions. For instance, in my experiments, CascadePSP improved the performance of CLIP-ES from 70.8 to around 77, whereas in the results presented, it only achieved a slight improvement on different settings, which is strange.
> > > >
> > > > Thank you for your comments. We would like to argue that  the improvements made by fully supervised methods such as CascadePSP and Segrefiner heavily rely on **the quality of coarse mask source**. The reason is that, during the training phase of these method, **the coarse mask generated is perturbed from the GT segmentations(as stated in Sect 4.2 Implementation Details in [5] and Sect 4.1 Implementation Details in [6])**, which is naturally of relatively high quality. When encountering a mask whose quality is markedly inferior than the ground truth, the performance gain of these methods may **degrade due to the supervised gap between the training and test input coarse masks**.
> > > >
> > > > We further make following clarification for CascadePSP detail implementation:
> > > >
> > > > 1. As the the mask quality of CLIP-ES is of good quality, the improvements may be significant compared with our SCLIP TSSS baseline. If the **mask quality is not as good, the overall improvement achieved by these methods may not be as significant**. As shown Table 4 (Table in Q1) in initial paper for a hard segmentation task on ADE20k, the improvement is also slight(42.2+**0.7**). Furthermore, the implementation code of CascadePSP on ADE20k have some **tuning parameters for segmentation logit** as shown in the code[1].
> > > >     ```python
> > > >         background_classes =[1,2,3,4,6,7,10,12,14,17,22,26,27,29,30,47,49,52,53,55,60,61,62,69,80,85,92,95,97,102,106,110,114,129,141]
> > > >         if this_class in background_classes:
> > > >             class_mask_prob = class_mask_prob * 0.51
> > > >     ```
> > > >
> > > > 2. For our implementation details, We utilized their evaluation code[2], primarily leveraging the **process_im_single_pass**[3] method to improve the base segmentation then utilizing the post combination[4].
> > > > ```python
> > > >    rmin, rmax, cmin, cmax = get_bb_position(class_seg)
> > > >    rmin, rmax, cmin, cmax = scale_bb_by(rmin, rmax, cmin, cmax, seg.shape[0], seg.shape[1], 0.25, 0.25)
> > > >    ...
> > > >    mask[rmin:rmax, cmin:cmax] = np.where(class_mask_prob>confidence[rmin:rmax, cmin:cmax],
> > > >                                                 this_class, mask[rmin:rmax, cmin:cmax])
> > > >    confidence[rmin:rmax, cmin:cmax] = np.maximum(confidence[rmin:rmax, cmin:cmax], class_mask_prob)
> > > > ```
> > > >
> > > > for multi-class processing. **If you have better tuning parameters such as [1] or implementation details for VOC, please offer us for a better comparison**.
> > > >
> > > >
> > > >
> > > > [1]https://github.com/hkchengrex/CascadePSP/blob/83cc3b8783b595b2e47c75016f93654eaddb7412/eval_post_ade.py#L127
> > > >
> > > > [2] https://github.com/hkchengrex/CascadePSP.
> > > >
> > > > [3]https://github.com/hkchengrex/CascadePSP/blob/83cc3b8783b595b2e47c75016f93654eaddb7412/eval_helper.py#L176
> > > >
> > > > [4]https://github.com/hkchengrex/CascadePSP/blob/83cc3b8783b595b2e47c75016f93654eaddb7412/eval_post_ade.py#L114
> > > >
> > > > [5] CascadePSP: Toward Class-Agnostic and Very High-Resolution Segmentation via Global and Local Refinement. CVPR2020.
> > > >
> > > > [6] SegRefiner: Towards Model-Agnostic Segmentation Refinement with Discrete Diffusion Process. NeurIPs2023.

---

> > > > > ### Comment · Reviewer_a25d · 2024-11-26
> > > > >
> > > > > Thanks for the author's explanation.
> > > > >
> > > > > 1. I acknowledge that `the intrinsic generative priors in diffusion models could be properly utilized for segmentation tasks and a number of approaches are exploring how diffusion models can enhance segmentation tasks.`  However, these priors may not be well-suited for the mask refinement task, as their inherent lack of precision limits their ability to guide the accurate refinement. For instance, previous works like DatasetDiffusion leverage attention mechanisms within diffusion models to produce segmentation masks, but the resulting masks are noticeably coarse, further highlighting the inaccuracy of generative priors in this context.
> > > > >
> > > > > 2. I think there may be a misuse of CascadePSP in your reproduction, at least on the VOC dataset. For example, using the evaluation code [2] for VOC is inappropriate because the VOC dataset contains the category 255, which should be ignored during evaluation. A more reasonable approach would be as follows:
> > > > >
> > > > > - (1) Split the semantic mask into binary masks (valued 0 and 255) for each category in the mask;
> > > > > - (2) Refine each binary mask using CascadePSP;
> > > > > - (3) Merge the refined binary masks into a semantic mask;
> > > > > - (4) Evaluate the merged masks using the evaluation codes provided by baseline methods such as SCLIP or CLIP-ES.
> > > > >
> > > > > By following this process, CascadePSP can improve the mIoU by nearly 10% on the SCLIP-VOC validation set in my case. I suspect the unusual reproduction results may stem from the incorrect binarization process or an improper evaluation method, making the comparison unfair and less convincing. If I have misunderstood any details or made an error in my analysis, please feel free to let me know. I would greatly appreciate any clarification.

---

> > ### Author Response · Authors · 2024-11-24
> >
> > Dear Reviewer a25d,
> >
> > Thank you again for reviewing our work and for providing valuable feedback. We sincerely hope our responses have addressed your comments and clarified the points raised.  Please let us know if you have further comments or need more discussions.
> >
> >
> > Best regards,
> >
> > Authors of Submission 6302

---

> ### Author Response · Authors · 2024-11-26
>
> I am very curious about your pipeline. **Since in the CascadePSP paper, the highest improvement in IoU is only 5% IoU among all the datasets**, which is in a simpler segmentation task (with only one foreground per image). In more complex multi-class segmentation tasks, it seems that the **10% IoU improvements on SCLIP and the 7% IoU improvements on CLIP-ES** are unclear or not well explained. Could you please clarify how these improvements were obtained?  Did your **reproduction align with the original paper?** Could you share your code with an anonymous link?

---

> ### Comment · Reviewer_a25d · 2024-11-26
>
> As you mentioned, the original CascadePSP is designed to process the simple foreground-background case. To apply it to semantic segmentation with multi-class segmentation tasks, we need to first convert the semantic mask to several binary masks for each class (as the step 1 listed above). For example, if the coarse mask contains two classes `person` and `dog`, we need to first split it into two binary masks (valued 0 and 255) for each class and perform refinement based on them.
>
> Additionally, the coarse masks used in CascadePSP are generated by **fully supervised models** like Deeplab V3+, which are already of high quality, making the observed improvements less significant. In contrast, for WSSS, the coarse masks have more room for improvement, so achieving higher gains is reasonable. Furthermore, the metric used in CascadePSP is **image-wise IoU**, differing from the **class-wise mIoU** typically used in semantic segmentation tasks. To ensure consistency with baseline methods such as SCLIP and CLIP-ES, mIoU should be adopted instead.
>
> I am also very curious about the difference between our reproductions. I would appreciate it if the authors could correct me and share their reproduction details of CascadePSP on VOC.

---

> > ### Author Response · Authors · 2024-11-26
> >
> > We have followed the SCLIP and mIoU implementations from MMseg. We would like to know if it would be possible for you to **share your implementation codebase with us**. To help you understand our approach, we have prepared our code for SCLIP. If you notice any mistakes or areas for improvement, please don't hesitate to let us know. The preparation of our code should be followed as following steps:
> > 1. Download the SCLIP code and prepare the SCLIP environment and mmseg style dataset for voc
> >
> > 2. Prepare the CascadePSP through
> > ```
> > pip install segmentation-refinement
> > ```
> > and then put the pretrained weights (https://github.com/hkchengrex/CascadePSP/blob/master/docs/models.md) under the SCLIP folder.
> >
> > 3. Then substitute the core SCLIP code "clip_segmentor.py" with our implementation https://anonymous.4open.science/r/SCLIP_CascadePSP-4142/clip_segmentor.py
> >
> > 4. Running the VOC evaluation with:
> > ```
> > python eval.py --config configs/cfg_voc21.py
> > ```
> > *Note that the code is a little modification with our implementation and for the convenient to discuss the performance gap*

---

> > > ### Comment · Reviewer_a25d · 2024-11-26
> > >
> > > I think the  `get_bb_position()` and `scale_bb_by()` functions are strange and unnecessary for the multi-class segmentation case because we can directly set the merged mask as `mask[class_mask_prob>confidence]=cate_id`. Additionally, I realized my earlier mistake in confusing the ground truth mask with the predicted mask when extracting `cate_ids`. Based on your implementation, I modified the code as follows, which ultimately improved the mIoU from 59% to about 65% on val set.
> > >
> > > ```python
> > > if self.use_cpsp:#True
> > >     seg_pred = seg_logits.argmax(0, keepdim=False)
> > >     seg_pred = seg_pred.cpu().numpy().astype(np.uint8)
> > >     confidence = np.zeros_like(seg_pred) + 0.5
> > >
> > >     # change this into tensor from the
> > >     img = cv2.imread(data_samples[0].img_path)
> > >     mask = np.zeros_like(seg_pred)
> > >
> > >     for elab in np.unique(seg_pred).tolist():
> > >         if elab != 0: # not for background
> > >             # waiting to refine
> > >             # changing into the cascaded psp format
> > >             seg = ((seg_pred==elab).astype(np.uint8)*255)[:,:,np.newaxis]
> > >
> > >             output = self.refiner.refine(img, seg, fast=True, L=900)
> > >             class_mask_prob = output / 255
> > >
> > >             mask[class_mask_prob>confidence] = elab
> > >             confidence = np.maximum(confidence, class_mask_prob)
> > >
> > >     seg_pred = torch.from_numpy(mask).unsqueeze(0).cuda()
> > >
> > > else:
> > >     seg_pred = seg_logits.argmax(0, keepdim=True)
> > >     seg_pred[seg_logits.max(0, keepdim=True)[0] < self.prob_thd] = 0

---

> > > > ### Author Response · Authors · 2024-11-27
> > > >
> > > > Thank you for pointing out the extra implementation details. We have made some improvements to our code base, but not as high as you provided. In fact, the *get_bb_position* and *scale_bb_by* operations are the ones done in the original Cascade code base for evaluating the ADE20k as scene parsing.  We will keep checking the experiments on CascadePSP for a better comparison.
> > > >
> > > > As you mentioned in the earlier discussion, the prior from the generative model may currently be coarse to serve as a segmentation mask. However, as the **controllability of generative models continues to improve—a challenge that many works are actively addressing—the effectiveness and advantages of our paradigm are expected to significantly increase**(ControlNet experiments).  In the future, when the base generative model is affiliated with strong position controllability, our framework paves the way for grounding in a generative manner.
> > > >
> > > > Finally, are there any remaining concerns or unresolved issues?

---

> > > > > ### Comment · Reviewer_a25d · 2024-11-27
> > > > >
> > > > > I greatly appreciate the author's active participation in the discussion. I believe this paper still has considerable room for improvement, and therefore, I lean to maintain my original score.

---

### Official Review · Reviewer_ADSn · 2024-11-01

**Soundness:** 3
**Presentation:** 3
**Contribution:** 3
**Rating:** 6
**Confidence:** 4

**Summary:**

This paper presents an innovative diffusion-based approach to address the Inexact Segmentation (IS) task by leveraging the intrinsic generative priors inherent in Stable Diffusion. The authors propose a coarse-to-fine segmentation refinement strategy that exploits pattern discrepancies between original images and mask-conditional generated images. This is achieved through the establishment of semantic correspondence alignment and iterative updating of foreground probabilities. A key contribution of this work is the introduction of a semantic correspondence alignment methodology, which employs the pixel-wise Hausdorff distance as a discrepancy metric. The efficacy of the proposed method is demonstrated through extensive experimentation in both open-vocabulary and weakly supervised segmentation tasks. Notably, the approach consistently outperforms current state-of-the-art methods, showcasing substantial performance gains across various benchmarks.

**Strengths:**

1.The proposed G4SEG introduces an innovative training-free framework for inexact segmentation refinement using generative models. This approach represents a significant departure from previous discriminative-based and diffusion model (DM)-based training methods, offering a novel perspective in the field.

2. The concept of explicit mask projection and semantic correspondence alignment is particularly noteworthy. By ingeniously decoupling the mask refinement process from target image reconstruction, the authors present a unique solution to the segmentation refinement problem.

3. The method demonstrates consistent performance gains when incorporated with current state-of-the-art approaches in both open-vocabulary and weakly supervised segmentation tasks. This broad applicability underscores the effectiveness and versatility of the proposed technique.

**Weaknesses:**

1. The reliance on Stable Diffusion may potentially limit the model's efficiency compared to baseline methods, raising concerns about computational requirements and processing speed.

2. The improvements are most pronounced when applied to less accurate segmentation models, such as GroupVit, which is trained solely with text labels. When used with stronger base segmentation models (e.g., SCLIP and DiffSegmenter), the performance gains appear to be less substantial, potentially limiting the method's applicability across a broader range of models.

**Questions:**

1. How does the proposed method handle cases where the initial segmentation is significantly inaccurate or contains multiple errors? Is there a threshold of initial accuracy below which the refinement process becomes less effective or unreliable?

2. Given that the method leverages generative priors from Stable Diffusion, how sensitive is it to domain shifts or out-of-distribution images? Have the authors explored its performance on datasets or image types that differ significantly from those used in training Stable Diffusion?

3. Could this approach be extended or adapted to those fully-supervised methods that can already achieve accurate segmentation results, such as SegFormer and Mask2Former?

---

> ### Author Response · Authors · 2024-11-21
> **Response to Reviewer ADSn 1/2**
>
> ### W1
> >The reliance on Stable Diffusion may potentially limit the model's efficiency compared to baseline methods, raising concerns about computational requirements and processing speed.
>
> While there may currently be computational challenges, compared with the other segmentation refinement method using diffusion generation with multi-step inference [1], our method only **needs one step of model inference for a specific class**.
>
> Furthermore, the model computational cost of diffusion models could be **improved with such as pruning, quantization, or using lighter versions of Stable Diffusion**. We conduct our experiments with a slighter version of stable diffusion achieving comparable results.
>
> | Dif Version                           | mIoU |
> | ------------------------------------- | ---- |
> | CLIP-ES Baseline                      | 70.8 |
> | +G4Seg with small stable diffusion[2] | 71.8 |
> | +G4Seg with SD2.1                     | 72.0 |
> As a method that leverages the generative capabilities of diffusion models for post-processing, it is natural for our approach to incur additional computational overhead. However, after comparing the **computational latency of several mask refinement methods**, we found that our approach operates within **a reasonable range of computational cost**.
> | method           | Inference Speed |
> | ---------------- | --------------- |
> | +G4Seg           | +1.3s           |
> | +Densecrf        | +0.9s           |
> | +Slide Inference | +1.8s           |
> | +SegRefiner[1]   | +1.7s           |
>
> *SegRefiner is a fully supervised mask refinement method with diffusion process*
>
>
>
> ### W2 & Q3
> >The improvements are most pronounced when applied to less accurate segmentation models, such as GroupVit, which is trained solely with text labels. When used with stronger base segmentation models (e.g., SCLIP and DiffSegmenter), the performance gains appear to be less substantial, potentially limiting the method's applicability across a broader range of models.
> >Could this approach be extended or adapted to those fully-supervised methods that can already achieve accurate segmentation results, such as SegFormer and Mask2Former?
>
> To show the versability of our method, we have conduct experients under a fully supervised setting on ADE20k with three fully supervised segmentation approaches(SegFormer[4],Mask2Former[5]) with semantic segmentaion and panoptic segmentation(ODISE[6]).
>
> **Fully supervised results on ADE20k datasets**
>
> | Methods          | mIoU/PQ        |
> | ---------------- | -------------- |
> | SegFormer (B1)   | 42.2           |
> | +G4Seg           | **42.9(+0.7)** |
> | Mask2Former(R50) | 47.2           |
> | +G4Seg           | **47.8(+0.6)** |
> | ODISE(panoptic)  | 22.4           |
> | +G4Seg           | **23.0(+0.6)** |
>
> We have demonstrated the applicability of our method in fully supervised settings. Note that our method is applied **without training or finetuning with any pixel-level annotation**.
>
> **The reason**
> To further clarify the limited performance on SCLIP and DiffSegmenter comparing GroupViT, we make following explanation. Specifically, the knowledge for mask quality improvements comes from two components: Explicit Mask Injection, which **relies on the attention of the diffusion backbone and the CLIP embedding space for correspondence mapping**; DiffSegmentor, which relies solely on the attention of the diffusion backbone, while SCLIP depends on CLIP's discriminative capability. Therefore, both methods **share some overlap in terms of knowledge sources with ours**, which limits the extent of improvement. If we apply another mask injection(Seg Controlnet[3]) and embedding space(Diffusion Backbone Feature), the accuracy improvements of our pipeline would be more significant.
>
> | Method                          | COCO     |
> | ------------------------------- | -------- |
> | SCLIP + G4Seg                   | 30.9     |
> | SCLIP + G4Seg(Diffusion Feat)   | **31.3** |
> | DiffSegmenter+G4Seg             | 38.5     |
> | DiffSegmenter+G4Seg(Controlnet) | **39.6** |

---

> ### Author Response · Authors · 2024-11-21
> **Response to Reviewer ADSn 2/2**
>
> ### Q1
> >How does the proposed method handle cases where the initial segmentation is significantly inaccurate or contains multiple errors? Is there a threshold of initial accuracy below which the refinement process becomes less effective or unreliable?
>
> Thank you for the constructive comments. To show how sensitive the proposed method relying on the initial segmentation quality, since our approach is sample-wise, we performed stratification based on different quality levels of coarse segmentation and then calculated the mean IoU improvement for samples with different levels for the VOC dataset:
>
> **Table Improvement for Different Initial Performance**
> | IoU   Group | 0-40     | 40-80      | 80-100     |
> | ----------- | -------- | ---------- | ---------- |
> | proportion  | 56(3.4%) | 679(47.3%) | 237(49.3%) |
> | Improvement | +0.2     | +1.9       | +1.1       |
>
> When the initial mask quality is poor, the improvement of our method is also limited. The improvement from our method is most significant for initial IoU values between 40 and 80. This indicates that **our approach is particularly effective when the initial segmentation is already of reasonable quality**. When the initial segmentation is already nearly perfect(80-100), the improvement from our method becomes limited due to the bottleneck caused by errors inherent in the mask injection process.
>
>
> ### Q2
> >Given that the method leverages generative priors from Stable Diffusion, how sensitive is it to domain shifts or out-of-distribution images? Have the authors explored its performance on datasets or image types that differ significantly from those used in training Stable Diffusion?
>
> Thank you for your advice. To evaluate the performance of our method in **cross-domain settings**, we adopted a baseline[7] for nighttime semantic segmentation on NightCity-fine[8].
>
> | Methods  | mIoU       |
> | -------- | ---------- |
> | DP[7]    | 64.0       |
> | DP+G4Seg | 64.5(+0.5) |
>
> Our method remains effective under out-of-domain conditions.
>
>
>
> >[1] SegRefiner: Towards Model-Agnostic Segmentation Refinement with Discrete Diffusion Process.NeurIPs2023
>
> >[2] https://huggingface.co/OFA-Sys/small-stable-diffusion-v0
>
> >[3] https://huggingface.co/lllyasviel/sd-controlnet-seg
>
> >[4] SegFormer: Simple and Efficient Design for Semantic Segmentation with Transformers. NeurIPs2021
>
> >[5] Mask2Former: Masked-attention Mask Transformer for Universal Image Segmentation. CVPR2022
>
> >[6] ODISE: Open-Vocabulary Panoptic Segmentation with Text-to-Image Diffusion Models. CVPR2023
>
> >[7] Disentangle then Parse: Night-time Semantic Segmentation with Illumination Disentanglement. ICCV2023
>
> >[8] Night-time scene parsing with a large real dataset. TIP2021

---

> ### Author Response · Authors · 2024-11-24
>
> Dear Reviewer ADSn:
>
> Thank you again for taking the time to review our work and for providing valuable and insightful feedback. We sincerely hope the responses have clarified the points raised. If you have any additional questions or concerns, please do not hesitate to reach out—we would be happy to discuss them further.
>
> Best regards,
>
> Authors of Submission 6302

---

### Official Review · Reviewer_ervM · 2024-11-03

**Soundness:** 3
**Presentation:** 3
**Contribution:** 3
**Rating:** 6
**Confidence:** 5

**Summary:**

The paper presents G4Seg, a novel approach leveraging large-scale pretrained diffusion models for refining inexact segmentation without additional training. The method capitalizes on discrepancies between an original image and its generated counterpart conditioned on a coarse segmentation mask to refine segmentation results. By applying a semantic correspondence alignment and employing pixel-wise probability updates, G4Seg proposes a training-free, plug-and-play solution applicable to weakly supervised and text-supervised semantic segmentation tasks.

**Strengths:**

1. The proposal to use generative discrepancies from diffusion models for segmentation refinement is unique and well-explored, emphasizing the potential of generative models in traditionally discriminative tasks.

2. The method’s independence from training requirements is noteworthy, making it resource-efficient and broadly adaptable.

3. Comprehensive evaluation across standard datasets (PASCAL VOC12, PASCAL Context, MS COCO) demonstrates consistent performance improvements, achieving state-of-the-art results in certain scenarios.

**Weaknesses:**

1.  While results are promising, deeper comparisons with similar methods (e.g., training-free segmentation approaches) could be emphasized to highlight advantages and potential trade-offs more explicitly.

2. Although tested on popular benchmarks, it is not entirely clear how G4Seg would scale to other datasets.

3. While computational cost is discussed, there is limited exploration of scenarios where G4Seg may not perform optimally (e.g., highly cluttered or ambiguous images).

**Questions:**

Please see weaknesses

---

> ### Author Response · Authors · 2024-11-21
> **Response to Reviewer ervM 1/2**
>
> ### W1
> >While results are promising, deeper comparisons with similar methods (e.g., training-free segmentation approaches) could be emphasized to highlight advantages and potential trade-offs more explicitly.
>
> We have compared our method with other diffusion based segmentation approaches in Table 1 in the manuscript. Furthermore, detailed and further comparison with similar training free segmentation methods is shown in following table. We dicussed in terms of the **Generative Content, the strategy mimic generation and segmentation gap(Gen->Seg) and external discriminative assitance**.
>
> | Training-free Methods | Generative Content       | Gen->Seg                         | Discriminative assitance         |
> | --------------------- | ------------------------ | -------------------------------- | -------------------------------- |
> | OVDiff[1]                | Class conidtioned images | Cross/Self attention             | Discriminative feature prototype |
> | Freeda[2]                | Class conidtioned images | Cross/Self attention             | Discriminative feature prototype |
> | DiffSegmentor[3]         | None                     | Cross/Self attention             | None                             |
> | G4Seg                 | Mask conidtioned images  | Semantic correspondence updating | None                             |
>
> **Advantage**
> As the table shows, the DiffSegmentor only relies on the attention mechanism in the diffusion backbone as the clue to the target mask, which does not fully excavate the generation prior of the diffusion model. The OVDiff and Freeda utilize many generated images with specific classes and obtain the discriminative prototype of the class, where the **prototype is retrieved based on the region of interest from cross/Self-attention aggregation**. Due to the strong external discriminative assistance, it is challenging to determine **whether the generative capacity of the diffusion model contributes** to the performance. Our method aims to **fully exploit the generative prior for a discriminative task**, specifically inexact segmentation, by adopting a GPT-like approach to solve the discriminative task in a **generative manner without overly relying on extra assistance**.
>
> **Potential trade-offs**
>
> Avoiding over-reliance on external discriminative features highlights the generative capacity of diffusion models but may sacrifice performance in challenging segmentation tasks. Furthermore, solving a discriminative task in a generative manner might struggle with tasks requiring precise boundary delineation, which are typically better handled by explicit discriminative features or accurate annotations.
>
> We have added these discussions in the revisions of Appendix M.
>
>
> ### W2
> >Although tested on popular benchmarks, it is not entirely clear how G4Seg would scale to other datasets.
>
> We extended our method to two scenarios: one involving cross-domain datasets and another featuring datasets with a larger number of categories.
>
> We firstly evaluate the performance of our method on a **cross-domain setting** and adopt a baseline[7] for nighttime semantic segmentation on NightCity-fine[8].
>
> | Methods  | mIoU       |
> | -------- | ---------- |
> | DP[7]    | 64.0       |
> | DP+G4Seg | 64.5(+0.5) |
>
> We further conduct our method on ADE20k with three **fully supervised segmentation approaches**(SegFormer[4],Mask2Former[5]) with semantic segmentation and panoptic segmentation(ODISE[6]).
>
> **Fully supervised results on ADE20k datasets**
>
> | Methods             | mIoU/PQ        |
> | ------------------- | -------------- |
> | SegFormer[4] (B1)   | 42.2           |
> | +G4Seg              | **42.9(+0.7)** |
> | Mask2Former[5] (R50) | 47.2           |
> | +G4Seg              | **47.8(+0.6)** |
> | ODISE[6] (panoptic)  | 22.4           |
> | +G4Seg              | **23.0(+0.6)** |
>
> We have added these experiments to our revision with Appendix H.

---

> ### Author Response · Authors · 2024-11-21
> **Response to Reviewer ervM 2/2**
>
> ### W3
> >While computational cost is discussed, there is limited exploration of scenarios where G4Seg may not perform optimally (e.g., highly cluttered or ambiguous images).
>
> Thank you for the constructive comments. To show how sensitive the proposed method relying on the initial segmentation quality, since our approach is sample-wise, we stratified the samples based on different quality levels of coarse segmentation and calculated the **mean IoU improvement for each group according to their initial mask quality**, as measured by the per-sample IoU metric on the VOC dataset:
>
> **Table Improvement for Different Initial Performance**
> | Initial Mask Quality(IoU range) | 0-40 | 40-80 | 80-100 |
> | ------| -------- | ---- | ----- |
> | # samples(#/# total samples)         | 56(3.4%) | 679(47.3%) | 237(49.3%) |
> | Improvement          | +0.2     | +1.9       | +1.1       |
>
> When the initial mask quality is poor, the improvement of our method is also limited. The improvement from our method is most significant for initial IoU values between 40 and 80. This indicates that **our approach is particularly effective when the initial segmentation is already of reasonable quality**. When the initial segmentation is already nearly perfect(80-100), the improvement from our method becomes limited due to the bottleneck caused by errors inherent in the mask injection process.
>
> >[1] Diffusion models for zero-shot open-vocabulary segmentation. arXiv:2306.09316.
>
> >[2] Training-Free Open-Vocabulary Segmentation with Offline Diffusion-Augmented Prototype Generation. CVPR2024
>
> >[3] Diffusion model is secretly a training-free open vocabulary semantic segmenter. arXiv:2309.02773
>
> >[4] SegFormer: Simple and Efficient Design for Semantic Segmentation with Transformers. NeurIPs2021
>
> >[5] Mask2Former: Masked-attention Mask Transformer for Universal Image Segmentation. CVPR2022
>
> >[6] ODISE: Open-Vocabulary Panoptic Segmentation with Text-to-Image Diffusion Models. CVPR2023
>
> >[7] Disentangle then Parse: Night-time Semantic Segmentation with Illumination Disentanglement. ICCV2023
>
> >[8] Night-time scene parsing with a large real dataset. TIP2021

---

> ### Author Response · Authors · 2024-11-24
>
> Dear Reviewer ervM:
>
> Thanks for reviewing our work and providing insightful comments. We hope our answers have helped to clarify the points discussed. Please let us know if you have further comments or need more discussions.
>
> Best regards,
>
> Authors of Submission 6302

---

> > ### Comment · Reviewer_ervM · 2024-11-26
> > **Final Rating**
> >
> > Thank you for your response. The authors have addressed my concerns, and therefore, I have a positive view of this paper.

---

> > > ### Author Response · Authors · 2024-11-26
> > >
> > > Thank you again for your thoughtful review and valuable feedback. We are pleased to hear that the concerns have been addressed, and we appreciate your positive evaluation.

---

### Official Review · Reviewer_EypU · 2024-11-04

**Soundness:** 2
**Presentation:** 2
**Contribution:** 2
**Rating:** 5
**Confidence:** 4

**Summary:**

This paper proposes an image segmentation method using a pretrained text-to-image diffusion model, specifically Stable Diffusion 2.1. Given a text prompt with a class label, which conditions the image generation in UNet layers, it computes a semantic difference between the original image and a mask-conditional generated image after a one-step denoising step. This discrepancy, which is computed by Hausdorff distance, is employed to update the foreground probability of each pixel. Initial coarse segmentation follows the settings in the selected baselines.

**Strengths:**

This paper provides a training-free method for open vocabulary single-class segmentation.

**Weaknesses:**

I carefully reviewed the paper. Rather than an extensive list of less critical and debatable issues, I will concentrate on the most significant concerns that influence my rating. My focus will be on the following three limitations:

1) The reliance on a specific class label in the text prompt limits this method to only a single category for a given image. It is not clear how this method extends multiple classes. Running the proposed method multiple times with multiple yet separate class labels would face the problem of post-consolidating, most likely handling contrasting segmentation results outside the diffusion steps, which seems non-optimal.

2) The main idea is that, given an imperfect segmentation mask, the generated image will have a discrepancy from the original image. This drives the intuition given in the paper that with a more accurate segmentation mask, the probability of generating the original image is more likely to be maximized. How this is implemented is described in 3.2.2 and Appendix E. Some explanation is given 317-323; however, the explanation also highlights the reliance on a good initial segmentation mask. The ablation study does not present how sensitive the proposed method is to the initial segmentation accuracy.

3) The segmentation mIoU improvement is marginal. Besides, the comparisons with stronger SOTA are missing OVAM (CVPR 2024) and CoDe (CVPR 2024). DeOP (ICCV 2023), which could be another interesting baseline.

I am open to reconsidering my rating based on a satisfactory rebuttal from the authors.

**Questions:**

How could this method extend to other UNet-based text-to-image diffusion models beyond SD2.1 (such as Wurstchen, SSD-1B, SDXL, SD1.5, etc.) and their distilled versions (such as single or reduced step variants, LCM versions)?

---

> ### Author Response · Authors · 2024-11-21
> **Response to Reviewer EypU 1/2**
>
> ### W1
> >The reliance on a specific class label in the text prompt limits this method to only a single category for a given image. It is not clear how this method extends multiple classes. Running the proposed method multiple times with multiple yet separate class labels would face the problem of post-consolidating, most likely handling contrasting segmentation results outside the diffusion steps, which seems non-optimal.
>
> **Multi-class Extension**
> Sorry for the misleading. Our current approach does indeed involve running the process multiple times with separate class labels in order to obtain results for different categories. For each specific class, We **treat all other segments as background and update the current segment logit independently**. Then the final segmentation is refined with updated logit after normalization. We have added these illustrations in the paper in Sec 4.1.
>
> **Non-optimal post-consolidating**
> We did not incorporate contrasting segmentations of  different classes within the diffusion step for the following two reasons:
> -  The challenge of handling the composed injection of different classes in a single pass. We did a validation experiment for **injection with multi-classes with concatenation of different class names as prompt and restrict their corresponding cross-attention and self-attention**. The results are presented in the following table. Performing such multi-class injection significantly degrades the overall performance, as the interference among classes might affect the details in the generation.
> - The separation of the generation and updating processes: We have modularized and decoupled the mask injection generation from the segmentation results updating process. This design enhances the flexibility of our pipeline, allowing for the substitution of stronger mask injection modules or more advanced updating strategies, thereby facilitating further exploration in future work.
>
>
> | Methods     | mIoU on VOC |
> | ----------- | ----------- |
> | multi-pass  | 72.0        |
> | single-pass | 70.9        |
>
>
>
>
> ### W2
> > The main idea is that, given an imperfect segmentation mask, the generated image will have a discrepancy from the original image. This drives the intuition given in the paper that with a more accurate segmentation mask, the probability of generating the original image is more likely to be maximized. How this is implemented is described in 3.2.2 and Appendix E. Some explanation is given 317-323; however, the explanation also highlights the reliance on a good initial segmentation mask. The ablation study does not present how sensitive the proposed method is to the initial segmentation accuracy.
>
>
> Thank you for the constructive comments. To evaluate the sensitivity of the proposed method to the quality of the initial segmentation, we stratified the samples into groups based on different levels of coarse segmentation quality. Since our approach is sample-wise, we can directly calculate **the mean IoU improvement for each group according to their initial mask quality**, as measured by the per-sample IoU metric on the VOC dataset:
>
> **Table Improvement for different initial performance**
> | Initial Mask Quality(IoU range) | 0-40     | 40-80      | 80-100     |
> | -------------------- | -------- | ---------- | ---------- |
> | # samples(#/# total samples)         | 56(3.4%) | 679(47.3%) | 237(49.3%) |
> | Improvement          | +0.2     | +1.9       | +1.1       |
>
> According to the above table, when the initial mask quality is poor, the improvement of our method is also limited. The improvement from our method is most significant for initial IoU values between 40 and 80. This indicates that **our approach is particularly effective when the initial segmentation is already of reasonable quality**. When the initial segmentation is already nearly perfect(80-100), the improvement from our method becomes limited due to the bottleneck caused by errors inherent in the mask injection process. We have added this into the revision in Sect 4.2.

---

> ### Author Response · Authors · 2024-11-21
> **Response to Reviewer EypU 2/2**
>
> ### W3
> > The segmentation mIoU improvement is marginal. Besides, the comparisons with stronger SOTA are missing OVAM (CVPR 2024) and CoDe (CVPR 2024). DeOP (ICCV 2023), which could be another interesting baseline.
>
> We sincerely thank the comments, particularly regarding the expansion of the experimental dimensions.
> **Marginal Improvements**:
> We argue that G4Seg is a **purely generative prior-based attempt at segmentation refinement**, which only leverages a diffusion model pretrained with a generation target. This framework does **not rely on any pixel-level annotations or extra fine-tuning**.
> In Sect. 3.3.1, we introduced a mask conditioning method, which is based on cross-attention and self-attention. These attention-based generation methods do not perform well in a mask-conditioned generation. If we adopt a stronger mask conditioning method, such as ControlNet[1], the performance would significantly improve, as shown in the following Table:
> | Method                          | CLIP-ES VOC | SCLIP CoCo | SCLIP Pascal Context |
> | ------------------------------- | ----------- | ---------- | ------------------- |
> | G4Seg+EMI(Attn Injection)       | 72.0        | 30.9       | 31.3                |
> | G4Seg+EMI(ControlNet Injection) | 74.1        | 33.1       | 33.8                |
>
> We do not adopt such a stronger mask injection strategy because it would violate the "inexact segmentation" setting in our approach. We have added these discussion to Appendix F.
>
> **Other Baselines**
> Sorry, we couldn't identify which paper "CoDe" refers to. Please help give us more details about "CoDe" and we will include it into the results and discussion. For the present, we first clarify the remaining two papers:
>
> - OVAM[2]: OVAM uses **manually annotated masks of generated images** to update token embeddings, which are then used to generate more images and corrected cross-attention-based pseudo masks. Strictly speaking, this method is not an "inexact segmentation" approach. Therefore, we categorize this method as a semi-supervised/fully supervised approach.
>
> - DeOP[3]: DeOP is inherently a fully supervised method, so we also categorize it as fully supervised approach.
>
>
> **Table Results on semi-supervised/fully supervised methods.**
> | Methods | VOC             | Context        |
> | ------- | --------------- | -------------- |
> | OVAM[2]    | 61.2            | 28.3           |
> | +G4Seg  | **62.1(+0.9)**  | **28.9(+0.6)** |
> | DeOP[3]    | 91.7            | 48.8           |
> | +G4Seg  | **92.1 (+0.4)** | **49.3(+0.5)** |
>
> *Note that the OVAM is evaluated with cross attention pseudo mask generated with text token paper offered. DeOp use the pretrained weight provided with GitHub repository.*
>
> We have added these discussions to Appendix G and H.
>
> ### Q1.
> > How could this method extend to other UNet-based text-to-image diffusion models beyond SD2.1 (such as Wurstchen, SSD-1B, SDXL, SD1.5, etc.) and their distilled versions (such as single or reduced step variants, LCM versions)?
>
> **How to extend**
> Our pipeline contains two step, Explicit Mask Injection and Semantic Correspondence Alignment after the generation. EMI is based on the backbone attention perturbation.
>
> SD1.5, SD2.1, LCM, and SDXL (with SSD-1B as a lighter distilled version of SDXL) **share largely similar U-Net backbone architectures, incorporating cross-attention and self-attention layers**. Consequently, the **EMI step is executed in a nearly identical manner** across these models. Then after the generation, the SCA step remains the same.
>
> Würstchen includes a **semantic compressor**, which introduces uncertainties regarding its impact on our pipeline. As a result, directly utilizing it may present certain challenges, requiring further investigation and analysis to ensure its compatibility and effectiveness.
>
>
> **Experimental Results**
> We have compared the results with SD1.5, SD2.1, SDXL and LCM.
> | Dif Version | mIoU |
> | ----------- | ---- |
> | SD1.5       | 71.8 |
> | SD2.1       | 72.0 |
> | SDXL        | 72.0 |
> | LCM         | 72.1 |
>
> As can be seen from the Table, the version of pretrained diffusion model has no obvious impact on the segmentation results. We appreciate the reviewer's comments and have added this to the revision as in Appendix L.
>
>
> ### Reference
> >[1] https://huggingface.co/lllyasviel/sd-controlnet-seg
>
> >[2] Open-Vocabulary Attention Maps with Token Optimization for Semantic Segmentation in Diffusion Models.CVPR2024
>
> >[3] Open Vocabulary Semantic Segmentation with Decoupled One-Pass Network. ICCV2023

---

> ### Author Response · Authors · 2024-11-24
>
> Dear Reviewer EypU:
>
> Thanks for reviewing our work and providing insightful comments. We hope our answers have addressed your concerns. Please let us know if you have further comments or need more discussions.
>
> Best regards,
>
> Authors of Submission 6302

---

### Author Response · Authors · 2024-11-21

We sincerely thank the reviewers for their thoughtful feedback and efforts in evaluating our work. Your valuable suggestions have greatly helped us improve the quality of our paper. We also deeply appreciate the substantial efforts invested by all the ACs and PCs in reviewing our work at the same time.



We are inspired by the encouragement from the reviewers, who evaluate our work as **resource-efficient**(ervM), **broadly adaptable**(ervM,XTz2) and **offering novel perspective**(ADSn). In response to the comments and concerns raised by the reviewers, before providing detailed responses to each reviewer individually, we also summarize the common main points here:



- **Reliance on the quality of coarse mask before refinement**.
We have conducted comprehensive experiments of our methods regarding the initial segmentation mask quality. The results indicate that our method shows limited improvement when the quality of the initial mask is very low. However, we want to emphasize that **this is a common limitation for many segmentation post-processing methods**. Moreover, by employing more advanced mask injection techniques, the improvement of our method can be significantly enhanced. More detailed discussion and experiments could be found in the response to reviewer EypU and XTz2.


- **Comparison with other segmentation refinement methods**.
We have added comparisons between our method and other mask refinement algorithms to demonstrate the effectiveness of our approach in improving inexact segmentation. Here, fundamentally different from both pixel-level supervised methods (CascadePSP/SegRefiner) and image prior-based methods (Dense CRF), **the knowledge source G4Seg relies on to enhance segmentation performance is the generative capacity of pre-trained diffusion model and the pattern discrepancy between the original and mask-conditioning reconstructed images**, making it **capable of achieving even higher performance when combined with other methods**, as shown in the following table. More detailed discussion could be found in response to reviewers XTz2 and a25d.



- **Computational cost concerns**.
Our approach is a segmentation post-processing method involving a pre-trained diffusion model. First, the computational cost of our method is **within a reasonable range compared to other segmentation post-processing methods**. Second, in scenarios requiring rapid application, the pre-trained diffusion model can be lightweight through techniques such as **pruning or distilling into a smaller model**.  More detailed discussion and experiments could be found in the response to reviewer ADSn and ervM.

| Methods             | SCLIP VOC      | SCLIP Context  | MaskCLIP VOC   | MaskCLIP Context |
| ------------------- | -------------- | -------------- | -------------- | ---------------- |
| Baseline            | 59.1           | 30.4           | 38.8           | 23.6             |
| +G4Seg              | 59.8(+0.7)     | 31.3(+0.9)     | 39.4(+0.6)     | 24.1(+0.5)       |
| +SegRefiner         | 59.3(+0.2)     | 30.7(+0.3)     | 39.1(+0.3)     | 23.9(+0.3)       |
| +CascadePSP         | 59.5(+0.4)     | 30.9(+0.5)     | 39.2(+0.4)     | 23.8(+0.2)       |
| +Densecrf           | 60.9(+1.8)     | 31.2(+0.8)     | 39.9(+1.1)     | 24.2(+0.6)       |
| +G4Seg + CascadePSP | 60.1(+1.0)     | 31.6(+1.2)     | 39.5(+0.7)     | 24.3(+0.7)       |
| +**G4Seg+Densecrf** | **62.1(+3.0)** | **32.0(+1.6)** | **40.1(+1.3)** | **24.6(+1.0)**   |

---

### Note · Authors · 2024-12-05

I have read and agree with the venue's withdrawal policy on behalf of myself and my co-authors.